# When Does Optimizing a Proper Loss Yield Calibration?

**Jarosław Błasiok**
Columbia University
jb4451@columbia.edu

**Parikshit Gopalan**
Apple
parik.g@gmail.com

**Lunjia Hu**
Stanford University
lunjia@stanford.edu

**Preetum Nakkiran**
Apple
preetum.nakkiran@gmail.com

## Abstract

Optimizing proper loss functions is popularly believed to yield predictors with good calibration properties; the intuition being that for such losses, the global optimum is to predict the ground-truth probabilities, which is indeed calibrated. However, typical machine learning models are trained to approximately minimize loss over restricted families of predictors, that are unlikely to contain the ground truth. Under what circumstances does optimizing proper loss over a restricted family yield calibrated models? What precise calibration guarantees does it give? In this work, we provide a rigorous answer to these questions. We replace the global optimality with a local optimality condition stipulating that the (proper) loss of the predictor cannot be reduced much by post-processing its predictions with a certain family of Lipschitz functions. We show that any predictor with this local optimality satisfies smooth calibration as defined in Kakade and Foster [2008], Błasiok et al. [2023b]. Local optimality is plausibly satisfied by well-trained DNNs, which suggests an explanation for why they are calibrated from proper loss minimization alone. Finally, we show that the connection between local optimality and calibration error goes both ways: nearly calibrated predictors are also nearly locally optimal.

## 1 Introduction

In supervised prediction with binary labels, two basic criteria by which we judge the quality of a predictor are accuracy and calibration. Given samples from a distribution $\mathcal{D}$ on $\mathcal{X} \times \{0, 1\}$ that corresponds to points $x$ from $\mathcal{X}$ with binary labels $y \in \{0, 1\}$, we wish to learn a predictor $f$ that assigns each $x$ a probability $f(x) \in [0, 1]$ that the label is 1. Informally, accuracy measures how close the predictor $f$ is to the ground-truth $f^*(x) = \mathbb{E}[y|x]$.[1] Calibration [Dawid, 1982, Foster and Vohra, 1998] is an interpretability notion originating from the literature on forecasting, which stipulates that the predictions of our model be meaningful as probabilities. For instance, on all points where $f(x) = 0.4$, calibration requires the true label to be 1 about $40\%$ of the time. Calibration and accuracy are complementary notions: a reasonably accurate predictor (which is still far from optimal) need not be calibrated, and calibrated predictors (e.g. predicting the average) can have poor accuracy. The notions are complementary but not necessarily conflicting; for example, the ground truth itself is both optimally accurate and perfectly calibrated.

Proper losses are central to the quest for accurate approximations of the ground truth. Informally a proper loss ensures that under binary labels $y$ drawn from the Bernoulli distribution with parameter $v^*$,

---

[1]Here we focus on the accuracy of the predictor $f$ as opposed to the accuracy of a classifier induced by $f$.

the expected loss $\mathbb{E}[\ell(y, v)]$ is minimized at $v = v^*$ itself. Familiar examples include the squared loss $\ell_{\mathsf{sq}}(y, v) = (y - v)^2$ and the cross-entropy loss $\ell_{\mathsf{xent}}(y, v) = y \log(1/v) + (1 - y) \log(1/(1 - v))$. A key property of proper losses is that over the space of all predictors, the expected loss $\mathbb{E}_{\mathcal{D}}[\ell(y, f(x))]$ is minimized by the ground truth $f^*$ which also gives calibration. However minimizing over all predictors is infeasible. Hence typical machine learning deviates from this ideal in many ways: we restrict ourselves to models of bounded capacity such as decision trees or neural nets of a given depth and architecture, and we optimize loss using algorithms like SGD, which are only expected to find approximate local minima in the parameter space. What calibration guarantees transfer to this more restricted but realistic setting?

**The State of Calibration & Proper Losses.** There is a folklore belief in some quarters that optimizing a proper loss, even in the restricted setting, will produce a calibrated predictor. Indeed, the user guide for scikit-learn [Pedregosa et al., 2011] claims that *LogisticRegression returns well calibrated predictions by default as it directly optimizes Log loss*[2]. Before going further, we should point out that this statement in its full generality is just not true: **for general function families $\mathcal{C}$, minimizing proper loss (even globally) over $\mathcal{C}$ might not result in a calibrated predictor**. Even if the class $\mathcal{C}$ contains perfectly calibrated predictors, the predictor found by loss minimization need not be (even close to) calibrated. A simple example showing that even logistic regression can fail to give calibrated predictors is provided in Appendix B.

Other papers suggest a close relation between minimizing a proper loss and calibration, but stop short of formalizing or quantifying the relationship. For example, in deep learning, Lakshminarayanan et al. [2017] states *calibration can be incentivised by proper scoring rules*. There is also a large body of work on post-hoc recalibration methods. The recipe here is to first train a predictor via loss minimization over the training set, then compose it with a simple post-processing function chosen to optimize cross-entropy on a holdout set [Platt, 1999, Zadrozny and Elkan, 2002]. See for instance Platt scaling [Platt, 1999], where the post-processing functions are sigmoids, and which continues to be used in deep learning (e.g. for temperature scaling as in Guo et al. [2017]). Google's data science practitioner's guide recommends that, for recalibration methods, *the calibration [objective] function should minimize a strictly proper scoring rule* [Richardson and Pospisil, 2021]. In short, these works suggest recalibration by minimizing a proper loss, sometimes in conjunction with a simple family of post-processing functions. However, there are no rigorous bounds on the calibration error using these methods, nor is there justification for a particular choice of post-processing functions.

Yet despite the lack of rigorous bounds, there are strong hints from practice that in certain settings, optimizing a proper-loss does indeed yield calibrated predictors. Perhaps the strongest evidence comes from the newest generation of deep neural networks (DNNs). These networks are trained to minimize a proper loss (usually cross-entropy) using SGD or variants, and are typically quite far from the ground truth, yet they turn out to be surprisingly well-calibrated. This empirical phenomenon occurs in both modern image classifiers [Minderer et al., 2021, Hendrycks* et al., 2020], and language models [Desai and Durrett, 2020, OpenAI, 2023]. The situation suggests that there is a key theoretical piece missing in our understanding of calibration, to capture why models can be so well-calibrated "out-of-the-box." This theory must be nuanced: not all ways of optimizing a proper loss with DNNs yields calibration. For example, the previous generation of image classifiers were poorly calibrated [Guo et al., 2017], though the much smaller networks before them were well-calibrated [Niculescu-Mizil and Caruana, 2005]. Whether DNNs are calibrated, then depends on the architecture, distribution, and training algorithm— and our theory must be compatible with this.

In summary, prior work suggests a close but complicated relationship between minimizing proper loss and calibration. On one hand, for simple models like logistic regression, proper loss minimization does not guarantee calibration. On the other hand, for DNNs, certain ways (but not all ways) of optimizing a proper loss appears to yield well-calibrated predictors. The goal of our work is to analyze this phenomena from a theoretical perspective.[3] Our motivating question is: **What minimal conditions on model family and training procedure guarantee that optimizing for a proper loss provably yields small calibration error?**

---

[2] `https://scikit-learn.org/stable/modules/calibration.html`. Log-loss is $\ell_{\mathsf{xent}}$ in our notation.

[3] For simplicity, we avoid generalization concerns and consider the setting where we minimize loss directly on the population distribution. This is a reasonable simplification in many learning settings with sufficient samples, such as the 1-epoch training of many Large-Language-Models.

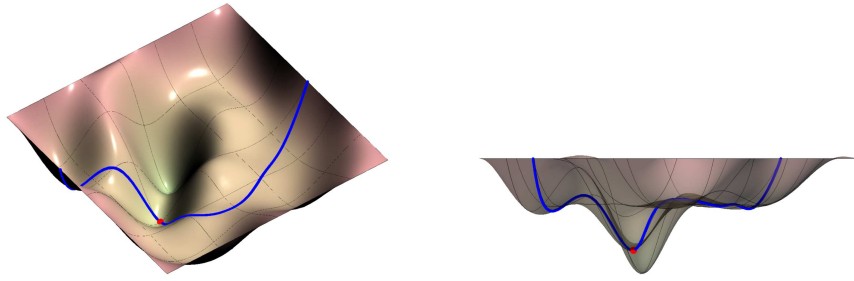

Figure 1: Schematic depiction of the loss landscape of a proper loss on the population distribution. The red dot represents a well-calibrated predictor $f$. The blue curve represents all predictors $\{\kappa \circ f : \kappa \in K_\ell\}$ obtained by admissible post-processing of $f$. Since the proper loss cannot be decreased by post-processing (the red dot is a minimal point on the blue curve), the predictor $f$ is well-calibrated, even though it is not a global minimum.

## 1.1 Our Contributions

The main contribution of this work is to identify a **local optimality condition** for a predictor that is both necessary and sufficient to guarantee calibration. The local optimality condition requires that the loss $\ell$ cannot be reduced significantly through post-processing using a family of functions $K_\ell$ that have certain Lipschitz-ness guarantees, where post-processing means applying a function $\kappa : [0, 1] \to [0, 1]$ to the output of the predictor. This condition is illustrated in Figure 1; it is distinct from the standard local optimality condition of vanishing gradients. We prove that a predictor satisfying this local optimality condition is smoothly calibrated in the sense of Kakade and Foster [2008], Gopalan et al. [2022b], Błasiok et al. [2023b]. Smooth calibration is a *consistent* calibration measure that has several advantages over the commonly used Expected Calibration Error (ECE); see the discussion in Kakade and Foster [2008], Błasiok et al. [2023b]. Quantitatively for the case of the squared loss, we present a tight connection between smooth calibration error and the reduction in loss that is possible from post-processing with $K_\ell$. For other proper loss functions $\ell$, we give a tight connection between post-processing with $K_\ell$ and a calibration notion we call dual smooth calibration. The fact that the connection goes both ways implies that a predictor that does not satisfy our local optimality condition is far from calibrated, so we have indeed identified the minimal conditions needed to ensure calibration.

**Implications.** Heuristically, we believe these theoretical results shed light on why modern DNNs are often calibrated in practice. There are at least three possible mechanisms, at varying levels of formality.

First, informally: poor calibration implies that the test loss can be reduced noticeably by post-processing with a *simple* function. But simple post-processings can be represented by adding a few layers to the DNN— so we could plausibly improve the test loss by adding a few layers and re-training (or continuing training) with SGD. Thus, if our DNN has been trained to the point where such easy gains are not left on the table (as we expect of state-of-the-art models), then it is well-calibrated.

The second possible mechanism follows from one way of formalizing the above heuristic, yielding natural learning algorithms that provably achieve calibration. Specifically, consider Structural Risk Minimization (SRM): globally minimizing a proper loss plus a complexity regularizer, within some restricted function family. In Section 5, we prove SRM is guaranteed to produce a calibrated predictor, provided: (1) the function family is closed under post-processing and (2) the complexity regularizer grows only mildly under post-processing. If the informal "implicit regularization hypothesis" in deep learning is true, and SGD on DNNs is equivalent to SRM with an appropriate complexity-regularizer [Arora et al., 2019, Neyshabur et al., 2014, Neyshabur, 2017, Zhang et al., 2021], then our results imply DNNs are well-calibrated as long as the regularizer satisfies our fairly mild assumptions.

Finally, a third mechanism for calibration involves a heuristic assumption about the optimizer (SGD). Informally, for a sufficiently deep network, updating the network parameters from computing the function $f$ to computing the post-processed $(\kappa \circ f)$ may be a "simple" update for SGD. Thus, if it were possible to improve the loss via such a simple post-processing, then SGD would have already exploited this by the end of training.

These algorithms and intuitions also suggest a practical guidance: for calibration, one should optimize loss over function families that are capable of computing rich classes of post-processing functions. The universality properties of most DNN architectures imply they can compute a rich family of post-processing functions with small added depth [Cybenko, 1989, Lu et al., 2017, Yun et al., 2019, Zhou, 2020] but the family of functions used in logistic regression (with logits restricted to being affine in the features) cannot.

Our results are consistent with the calibration differences between current and previous generation models, when viewed through the lens of generalization. In both settings, models are expected to be locally-optimal with respect to *train loss* [Carrell et al., 2022], since this is the explicit optimization objective. Calibration, however, requires local-optimality with respect to *test loss*. Previous generation image classification models were trained to interpolate on small datasets, so optimizing the train loss was very different from optimizing the test loss [e.g. Guo et al., 2017, Mukhoti et al., 2020]. Current generation models, in contrast, are trained on massive datasets, where optimizing the train loss is effectively equivalent to optimizing the test loss— and so models are close to locally-optimal with respect to both, implying calibration.

**Organization.** We first give a technical overview of our results in Section 2. In Section 3 we prove our main result in the case of square-loss. Then in Section 4 we extend this result to a wide family of proper losses. Finally, in Section 5 we present natural algorithms which provably achieve calibration, as a consequence by our results. Additional related works are presented in Appendix A.

## 2 Overview

In this section, we present a formal but still high-level overview of our results. We first set some notation. All our probabilities and expectations are taken with respect to a distribution $\mathcal{D}$ over $\mathcal{X} \times \{0,1\}$. A predictor is a function $f : \mathcal{X} \to [0,1]$ that assigns probabilities to points, the ground-truth predictor is $f^*(x) = \mathbb{E}[y|x]$. A predictor is perfectly calibrated if the set $A = \{v \in [0,1] : \mathbb{E}[y|f(x) = v] \neq v\}$ has measure 0 (i.e. $\mathrm{Pr}_{\mathcal{D}}(f(x) \in A) = 0$). A post-processing function is a function $\kappa : [0,1] \to [0,1]$. A loss function is a function $\ell : \{0,1\} \times [0,1] \to \mathbb{R}$, where $\ell(y,v)$ represents the loss suffered when we predict $v \in [0,1]$ and the label is $y \in \{0,1\}$. Such a loss is *proper* if when $y \sim \mathrm{Ber}(v^*)$ is sampled from the Bernoulli distribution with parameter $v^*$, the expected loss $\mathbb{E}[\ell(y,v)]$ is minimized at $v = v^*$, and we say a loss is *strictly* proper if $v^*$ is the unique minimizer.

As a warm-up, we present a characterization of perfect calibration in terms of perfect local optimally over the space of all possible (non-Lipschitz) post-processing functions.

**Claim 2.1.** *For every strictly proper loss $\ell$, a predictor $f$ is perfectly calibrated iff for every post-processing function $\kappa$,*

$$\mathbb{E}_{\mathcal{D}}[\ell(y,f(x))] \leq \mathbb{E}_{\mathcal{D}}[\ell(y,\kappa(f(x)))]. \tag{1}$$

This is true, because if $f$ is perfectly calibrated, and $\kappa$ arbitrary, then after conditioning on arbitrary prediction $f(x)$ we have:

$$\mathbb{E}[\ell(y,f(x))|f(x) = v] = \mathbb{E}_{y \sim \mathrm{Ber}(v)}[\ell(y,v)] \leq \mathbb{E}_{y \sim \mathrm{Ber}(v)}[\ell(y,\kappa(v))],$$

and (1) follows by averaging over $v$. On the other hand if $f$ is not perfectly calibrated, we can improve the expected loss by taking $\kappa(v) := \mathbb{E}[y|f(x) = v]$ — by the definition of the strictly proper loss, for each $v$ we will have $\mathbb{E}[\ell(y,\kappa(f(x))|f(x) = v] \leq \mathbb{E}[\ell(y,f(x))|f(x) = v]$, and on a set of positive probability the inequality will be strict.

While we are not aware of this precise claim appearing previously, statements that are similar in spirit are known; for instance, by the seminal work of Foster and Vohra [1998] which characterizes calibration in terms of swap regret for the squared loss. In our language they show the equivalent Claim 2.1 for $\ell$ being a squared loss, and $\kappa$ restricted to be of the form $\kappa(v_0) := w_0$, $\kappa(v) := v$ for all $v \neq v_0$.

Claim 2.1 connects calibration to a local optimality condition over the space of all possible post-processing functions. If this is satisfied, it guarantees calibration even though the loss might be far from the global minimum. This sheds light on classes of predictors such as decision trees or branching programs, which are closed under composition with arbitrary post-processing functions,

since this amounts to relabelling the leaf nodes. It tells us that such models ought to be calibrated from proper loss minimization, as long as the optimization achieves a fairly weak notion of local optimality.

The next step is to ask whether DNNs could satisfy a similar property. Closure under arbitrary post-processing seems too strong a condition for DNNs to satisfy — we should not expect DNNs to be able to express arbitrarily discontinuous uni-variate post-processing functions. But we do not expect or require DNNs to be perfectly calibrated, only to be close to perfectly calibrated. Measuring the distance from calibration involves subtle challenges as highlighted in the recent work of Błasiok et al. [2023b]. This leads us to the following robust formulation which significantly generalizes Claim 2.1:

- Using the well known relation between proper losses and Bregman divergences [Savage, 1971, Gneiting and Raftery, 2007], we consider proper loss functions where the underlying convex functions satisfy certain differentiability and smoothness properties. This is a technical condition, but one that is satisfied by all proper loss functions commonly used in practice.

- We only allow post-processing functions that satisfy certain Lipschitzness properties. Such functions are simple enough to be implementable by DNNs of constant depth. Indeed we heuristically believe that state-of-the-art DNNs will automatically be near-optimal w.r.t. such post-processings, as mentioned in Section 1.1.

- We measure calibration distance using the notion of smooth calibration from Kakade and Foster [2008] and strengthenings of it in Gopalan et al. [2022b], Błasiok et al. [2023b]. Smooth calibration plays a central role in the consistent calibration measures framework of Błasiok et al. [2023b], where it is shown using duality that the smCE is linearly related with the Wasserstein distance to the nearest perfectly calibrated predictor [Błasiok et al., 2023b]. In practice, there exist sample-efficient and linear-time estimators which approximate the smooth calibration error within a quadratic factor [Błasiok et al., 2023b].

With these notions in place, we show that the *post-processing gap* of a predictor, which measures how much the loss can be reduced via certain Lipschitz post-processing functions provides both an upper and a lower bound on the smooth calibration error. This result gives us a robust and quantitative version of Claim 2.1. The main technical challenge in formulating this equivalence is that for general $\ell$, the class of post-processing functions that we allow and the class of smooth functions that we use to measure calibration error are both now dependent on the loss function $\ell$, they are Lipschitz over an appropriately defined dual space to the predictions defined using convex duality. In order to keep the statements simple, we state our results for the special case of squared loss and cross entropy loss below, defering the full statement to the technical sections.

**Squared loss.** For the squared loss $\ell_{\mathsf{sq}}(y, v) = (y - v)^2$, we define the post-processing gap $\mathsf{pGap}_{\mathcal{D}}(f)$ to be the difference between the expected squared loss $\mathbb{E}[\ell_{\mathsf{sq}}(y, f(x))]$ of a predictor $f$ and the minimum expected squared loss after we post-process $f(x) \mapsto f(x) + \eta(f(x))$ for some 1-Lipschitz $\eta$. The Brier score [Brier et al., 1950, Foster and Vohra, 1998] uses the squared loss of a predictor as a calibration measure; $\mathsf{pGap}$ can be viewed as a *differential* version of Brier score.

**Definition 2.2** (Post-processing gap)**.** *Let $K$ denote the family of all post-processing functions $\kappa : [0, 1] \to [0, 1]$ such that the update function $\eta(v) = \kappa(v) - v$ is 1-Lipschitz. For a predictor $f : \mathcal{X} \to [0, 1]$ and a distribution $\mathcal{D}$ over $\mathcal{X} \times \{0, 1\}$, we define the post-processing gap of $f$ w.r.t. $\mathcal{D}$ to be*

$$\mathsf{pGap}_{\mathcal{D}}(f) := \mathbb{E}_{(x,y)\sim\mathcal{D}}[\ell_{\mathsf{sq}}(y, f(x))] - \inf_{\kappa \in K} \mathbb{E}_{(x,y)\sim\mathcal{D}}[\ell_{\mathsf{sq}}(y, \kappa(f(x)))].$$

We define the smooth calibration error $\mathsf{smCE}_{\mathcal{D}}(f)$ as in Kakade and Foster [2008], Gopalan et al. [2022b], Błasiok et al. [2023b] to be the maximum correlation between $y - f(x)$ and $\eta(f(x))$ over all bounded 1-Lipschitz functions $\eta$. Smooth calibration is a *consistent* measure of calibration [Błasiok et al., 2023b]. It does not suffer the discontinuity problems of ECE [Kakade and Foster, 2008], and is known to be linearly related with the Wasserstein distance to the nearest perfectly calibrated predictor (Błasiok et al. [2023b]). In particular, it is 0 if and only if we have perfect calibration. We refer the reader to these papers for a detailed discussion of its merits.

**Definition 2.3** (Smooth calibration error)**.** *Let $H$ be the family of all 1-Lipschitz functions $\eta : [0, 1] \to [-1, 1]$. The smooth calibration error of a predictor $f : \mathcal{X} \to [0, 1]$ with respect to distribution $\mathcal{D}$*

*over $\mathcal{X} \times \{0, 1\}$ is defined as*

$$\mathsf{smCE}_{\mathcal{D}}(f) := \sup_{\eta \in H} \mathbb{E}_{(x,y)\sim\mathcal{D}}[(y - f(x))\eta(f(x))].$$

Our main result is a quadratic relationship between these two quantities:

**Theorem 2.4.** *For any predictor $f : \mathcal{X} \to [0, 1]$ and any distribution $\mathcal{D}$ over $\mathcal{X} \times \{0, 1\}$,*

$$\mathsf{smCE}_{\mathcal{D}}(f)^2 \le \mathsf{pGap}_{\mathcal{D}}(f) \le 2\,\mathsf{smCE}_{\mathcal{D}}(f).$$

In Appendix D we show that the constants in the inequality above are optimal.

**Cross-entropy loss.**   For the cross entropy loss $\ell_{\mathsf{xent}}(y, v) = -y \ln v - (1 - y) \ln(1 - v)$, we observe its close connection to the logistic loss $\ell^{(\psi)}(y, t) = \ln(1 + e^t) - yt$ given by the equation

$$\ell_{\mathsf{xent}}(y, v) = \ell^{(\psi)}(y, t) \tag{2}$$

where $t := \ln(v/(1 - v))$ is what we call the *dual prediction* corresponding to $v$. While a standard terminology for $t$ is *logit*, we say $t$ is the dual prediction because this notion generalizes to arbitrary proper loss functions. One can conversely obtain a prediction $v$ from its dual prediction $t$ by taking the sigmoid transformation: $v = \sigma(t) := 1/(1 + e^{-t})$. The superscript $\psi$ in the logistic loss $\ell^{(\psi)}$ can be understood to indicate its relationship to the dual prediction and the exact meaning is made clear in Section 4. Based on (2), optimizing the cross-entropy loss $\ell_{\mathsf{xent}}$ over predictions $v$ is equivalent to optimizing the logistic loss $\ell^{(\psi)}$ over dual predictions $t$.

Usually, a neural network that aims to minimize the cross-entropy loss has a last layer that computes the sigmoid $\sigma$ (the binary version of softmax), so the value computed by the network before the sigmoid (the "logit") is the dual prediction. If we want to enhance the neural network by adding more layers, these layers are typically inserted before the final sigmoid transformation. It is thus more natural to consider post-processings on the dual predictions (logits) rather than on the predictions themselves.

For a predictor $f$, we define the *dual post-processing gap* $\mathsf{pGap}^{(\psi,1/4)}(g)$ of its dual predictor $g(x) = \ln(f(x)/(1 - f(x)))$ to be the difference between the expected logistic loss $\mathbb{E}[\ell^{(\psi)}(y, g(x))]$ of $g$ and the minimum expected logistic loss after we post-process $g(x) \mapsto g(x) + \eta(g(x))$ for some 1-Lipschitz $\eta : \mathbb{R} \to [-4, 4]$, where the constant 4 comes from the fact that the logistic loss is $1/4$-smooth in $t$.

**Definition 2.5** (Dual post-processing gap for cross-entropy loss, special case of Definition 4.4). *Let $K$ denote the family of all post-processing functions $\kappa : \mathbb{R} \to \mathbb{R}$ such that the update function $\eta(t) := \kappa(t) - t$ is 1-Lipschitz and bounded $|\eta(t)| \le 4$. Let $\ell^{(\psi)}$ be the logistic loss. Let $\mathcal{D}$ be a distribution over $\mathcal{X} \times \{0, 1\}$. We define the* dual post-processing gap *of a function $g : \mathcal{X} \to \mathbb{R}$ to be*

$$\mathsf{pGap}_{\mathcal{D}}^{(\psi,1/4)}(g) := \mathbb{E}_{(x,y)\sim\mathcal{D}}\,\ell^{(\psi)}(y, g(x)) - \inf_{\kappa \in K} \mathbb{E}_{(x,y)\sim\mathcal{D}}\,\ell^{(\psi)}\big(y, \kappa(g(x))\big).$$

We define the *dual smooth calibration error* $\mathsf{smCE}^{(\psi,1/4)}(g)$ to be the maximum of $|\mathbb{E}[(y - f(x))\eta(g(x))]|$ over all $1/4$-Lipschitz functions $\eta : \mathbb{R} \to [-1, 1]$. Like with smooth calibration, it is 0 if and only if the predictor $f$ is perfectly calibrated.

**Definition 2.6** (Dual smooth calibration for cross-entropy loss, special case of Definition 4.5). *Let $H$ be the family of all $1/4$-Lipschitz functions $\eta : \mathbb{R} \to [-1, 1]$. For a function $g : \mathcal{X} \to \mathbb{R}$, define predictor $f : \mathcal{X} \to [0, 1]$ such that $f(x) = \sigma(g(x))$ for every $x \in \mathcal{X}$ where $\sigma$ is the sigmoid transformation. Let $\mathcal{D}$ be a distribution over $\mathcal{X} \times \{0, 1\}$. We define the* dual calibration error *of $g$ as*

$$\mathsf{smCE}_{\mathcal{D}}^{(\psi,1/4)}(g) := \sup_{\eta \in H} |\mathbb{E}_{(x,y)\sim\mathcal{D}}[(y - f(x))\eta(g(x))]|.$$

We show a result similar to Theorem 2.4 that the dual post-processing gap and the dual smooth calibration error are also quadratically related. Moreover, we show that a small dual smooth calibration error implies a small (standard) smooth calibration error.

**Corollary 2.7** (Corollary of Theorem 4.6 and Lemma 4.7). *Let $\mathsf{pGap}^{(\psi,1/4)}$ and $\mathsf{smCE}^{(\psi,1/4)}$ be defined as in Definition 2.5 and Definition 2.6 for the cross-entropy loss. For any function $g : \mathcal{X} \to \mathbb{R}$ and any distribution $\mathcal{D}$ over $\mathcal{X} \times \{0, 1\}$,*

$$2\,\mathsf{smCE}_{\mathcal{D}}^{(\psi,1/4)}(g)^2 \le \mathsf{pGap}_{\mathcal{D}}^{(\psi,1/4)}(g) \le 4\,\mathsf{smCE}_{\mathcal{D}}^{(\psi,1/4)}(g). \tag{3}$$

*Moreover, let predictor $f : \mathcal{X} \to [0, 1]$ be given by $f(x) = \sigma(g(x))$ for the sigmoid transformation $\sigma$. Its (standard) smooth calibration error $\mathsf{smCE}_{\mathcal{D}}(f)$ defined in Definition 2.3 satisfies*

$$\mathsf{smCE}_{\mathcal{D}}(f) \leq \mathsf{smCE}_{\mathcal{D}}^{(\psi, 1/4)}(g). \tag{4}$$

The constants in the corollary above are optimal as we show in Lemmas D.3 and D.4 in the appendix. Both results (3) and (4) generalize to a wide class of proper loss functions as we show in Section 4.

**Remark 2.8.** *In a subsequent work, Błasiok and Nakkiran [2023] proved the reverse direction of* (4) *for the cross-entropy loss:* $\mathsf{smCE}_{\mathcal{D}}(f) \geq \Omega\big(\mathsf{smCE}_{\mathcal{D}}^{(\psi, 1/4)}(g)^2\big)$. *Hence both* $\mathsf{smCE}^{(\psi, 1/4)}(g)$ *and* $\mathsf{pGap}^{(\psi, 1/4)}(g)$ *are* consistent calibration measures, *a notion introduced by Błasiok et al. [2023b].*

Our result shows that achieving a small dual post-processing gap when optimizing a cross-entropy is necessary and sufficient for good calibration guarantees. It sheds light on the examples where logistic regression fails to yield a calibrated predictor; in those instances the cross-entropy loss can be further reduced by some Lipschitz post-processing of the logit. This is intuitive because logistic regression only optimizes cross-entropy within the restricted class where the logit is a linear combination of the features plus a bias term, and this class is not closed under Lipschitz post-processing.

# 3 Calibration and Post-processing Gap for the Squared Loss

In this section we prove our main result Theorem 2.4 relating the smooth calibration error of a predictor to its post-processing gap with respect to the squared loss.

*Proof of Theorem 2.4.* We first prove the upper bound on $\mathsf{pGap}_{\mathcal{D}}(f)$. For any $\kappa \in K$ in the definition of $\mathsf{pGap}$ (Definition 2.2), there exists a 1-Lipschitz function $\eta : [0, 1] \to [-1, 1]$ such that $\kappa(v) = v + \eta(v)$ for every $v \in [0, 1]$. For the squared loss $\ell$,

$$\begin{aligned}
\mathbb{E}_{(x,y) \sim \mathcal{D}}[\ell(y, \kappa(f(x)))] &= \mathbb{E}[(y - f(x) - \eta(f(x)))^2] \\
&= \mathbb{E}[(y - f(x))^2] - 2\,\mathbb{E}[(y - f(x))\eta(f(x))] + \mathbb{E}[\eta(f(x))^2]. \tag{5}
\end{aligned}$$

The three terms on the right hand side satisfy

$$\begin{aligned}
\mathbb{E}[(y - f(x))^2] &= \mathbb{E}[\ell(y, f(x))], \\
\mathbb{E}[(y - f(x))\eta(f(x))] &\leq \mathsf{smCE}_{\mathcal{D}}(f), \\
\mathbb{E}[\eta(f(x))^2] &\geq 0.
\end{aligned}$$

Plugging these into (5), we get

$$\mathbb{E}[\ell(y, f(x))] - \mathbb{E}[\ell(y, \kappa(f(x)))] \leq 2\mathsf{smCE}_{\mathcal{D}}(f).$$

Since this inequality holds for any $\kappa \in K$, we get $\mathsf{pGap}_{\mathcal{D}}(f) \leq 2\mathsf{smCE}_{\mathcal{D}}(f)$ as desired.

Now we prove the lower bound on $\mathsf{pGap}_{\mathcal{D}}(f)$. For any 1-Lipschitz function $\eta : [0, 1] \to [-1, 1]$, define $\beta := \mathbb{E}[(y - f(x))\eta(f(x))] \in [-1, 1]$ and define post-processing $\kappa : [0, 1] \to [0, 1]$ such that

$$\kappa(v) = \mathsf{proj}_{[0,1]}(v + \beta\eta(v)) \quad \text{for every } v \in [0, 1],$$

where $\mathsf{proj}_{[0,1]}(u)$ is the value in $[0, 1]$ closest to $u$, i.e., $\mathsf{proj}_{[0,1]}(u) = \min(\max(u, 0), 1)$. By Lemma H.2, we have $\kappa \in K$. The expected squared loss after we apply the post-processing $\kappa$ can be bounded as follows:

$$\begin{aligned}
\mathbb{E}_{(x,y) \sim \mathcal{D}}[\ell(y, \kappa(f(x)))] &\leq \mathbb{E}[(y - f(x) - \beta\eta(f(x)))^2] \\
&= \mathbb{E}[(y - f(x))^2] - 2\beta\,\mathbb{E}[(y - f(x))\eta(f(x))] + \beta^2\,\mathbb{E}[\eta(f(x))^2] \\
&= \mathbb{E}[\ell(y, f(x))] - 2\beta^2 + \beta^2\,\mathbb{E}[\eta(f(x))^2] \\
&\leq \mathbb{E}[\ell(y, f(x))] - 2\beta^2 + \beta^2.
\end{aligned}$$

Re-arranging the inequality above, we have

$$\mathbb{E}[(y - f(x))\eta(f(x))]^2 = \beta^2 \leq \mathbb{E}[\ell(y, f(x))] - \mathbb{E}[\ell(y, \kappa(f(x)))] \leq \mathsf{pGap}_{\mathcal{D}}(f).$$

Since this inequality holds for any 1-Lipschitz function $\eta : [0, 1] \to [-1, 1]$, we get $\mathsf{smCE}_{\mathcal{D}}(f)^2 \leq \mathsf{pGap}_{\mathcal{D}}(f)$, as desired. □

In Appendix D we provide examples showing that the constants in Theorem 2.4 are optimal.

# 4 Generalization to Any Proper Loss

When we aim to minimize the squared loss, we have shown that achieving a small post-processing gap w.r.t. Lipschitz post-processings ensures a small smooth calibration error and vice versa. In this section, we extend this result to a wide class of *proper* loss functions including the popular cross-entropy loss.

**Definition 4.1.** *Let $V \subseteq [0,1]$ be a non-empty interval. We say a loss function $\ell : \{0,1\} \times V \to \mathbb{R}$ is* proper *if for every $v \in V$, it holds that $v \in \operatorname{argmin}_{v' \in V} \mathbb{E}_{y \sim \mathsf{Ber}(v)}[\ell(y, v')]$.*

One can easily verify that the squared loss $\ell(y, v) = (y - v)^2$ is a proper loss function over $V = [0,1]$, and the cross entropy loss $\ell(y, v) = -y \ln v - (1 - y) \ln(1 - v)$ is a proper loss function over $V = (0,1)$.

It turns out that if we directly replace the squared loss in Definition 2.2 with an arbitrary proper loss, we do not get a similar result as Theorem 2.4. In Appendix C we give a simple example where the post-processing gap w.r.t. the cross-entropy loss can be arbitrarily larger than the smooth calibration error. Thus new ideas are needed to extend Theorem 2.4 to general proper loss functions. We leverage a general theory involving correspondence between proper loss functions and convex functions [Shuford et al., 1966, Savage, 1971, Schervish, 1989, Buja et al., 2005]. We provide a detailed description of this theory in Appendix E. There we include a proof of Lemma E.4 which implies the following lemma:

**Lemma 4.2.** *Let $V \subseteq [0,1]$ be a non-empty interval. Let $\ell : \{0,1\} \times V \to \mathbb{R}$ be a proper loss function. For every $v \in V$, define $\mathsf{dual}(v) := \ell(0, v) - \ell(1, v)$. Then there exists a convex function $\psi : \mathbb{R} \to \mathbb{R}$ such that*

$$\ell(y, v) = \psi(\mathsf{dual}(v)) - y \, \mathsf{dual}(v) \quad \text{for every } y \in \{0,1\} \text{ and } v \in V. \tag{6}$$

*Moreover, if $\psi$ is differentiable, then $\nabla \psi(t) \in [0,1]$.*

Lemma 4.2 says that every proper loss induces a convex function $\psi$ and a mapping $\mathsf{dual} : V \to \mathbb{R}$ that maps a prediction $v \in V \subseteq [0,1]$ to its *dual prediction* $\mathsf{dual}(v) \in \mathbb{R}$. Our main theorem applies whenever the induced function $\psi$ is differentiable, and $\nabla \psi$ is $\lambda$-Lipschitz (equivalently $\psi$ is $\lambda$-smooth) — those are relatively mild conditions that are satisfied by all proper loss functions of interest.

The duality relationship between $v$ and $\mathsf{dual}(v)$ comes from the fact that each $(v, \mathsf{dual}(v))$ pair makes the Fenchel-Young divergence induced by a conjugate pair of convex functions $(\varphi, \psi)$ take its minimum value zero, as we show in Appendix E. Equation (6) expresses a proper loss as a function that depends on $\mathsf{dual}(v)$ rather than directly on $v$, and thus minimizing a proper loss over predictions $v$ is equivalent to minimizing a corresponding *dual loss* over dual predictions $t = \mathsf{dual}(v)$:

**Definition 4.3** (Dual loss). *For a function $\psi : \mathbb{R} \to \mathbb{R}$, we define a* dual loss *function $\ell^{(\psi)} : \{0,1\} \times \mathbb{R} \to \mathbb{R}$ such that*

$$\ell^{(\psi)}(y, t) = \psi(t) - yt \quad \text{for every } y \in \{0,1\} \text{ and } t \in \mathbb{R}.$$

*Consequently, if a loss function $\ell : \{0,1\} \times V \to \mathbb{R}$ satisfies (6) for some $V \subseteq [0,1]$ and $\mathsf{dual} : V \to \mathbb{R}$, then*

$$\ell(y, v) = \ell^{(\psi)}(y, \mathsf{dual}(v)) \quad \text{for every } y \in \{0,1\} \text{ and } v \in V. \tag{7}$$

The above definition of a dual loss function is essentially the definition of the Fenchel-Young loss in the literature [see e.g. Duchi et al., 2018, Blondel et al., 2020]. A loss function $\ell^{(\psi)}$ satisfying the relationship in (7) has been referred to as a *composite loss* [see e.g. Buja et al., 2005, Reid and Williamson, 2010].

For the cross-entropy loss $\ell(y, v) = -y \ln v - (1 - y) \ln(1 - v)$, the corresponding dual loss is the logistic loss $\ell^{(\psi)}(y, t) = \ln(1 + e^t) - yt$, and the relationship between a prediction $v \in (0,1)$ and its dual prediction $t = \mathsf{dual}(v) \in \mathbb{R}$ is given by $v = \sigma(t)$ for the sigmoid transformation $\sigma(t) = e^t/(1 + e^t)$.

For a predictor $f : \mathcal{X} \to V$, we define its dual post-processing gap by considering dual predictions $g(x) = \mathsf{dual}(f(x))$ w.r.t. the dual loss $\ell^{(\psi)}$ as follows:

**Definition 4.4** (Dual post-processing gap). *For $\lambda > 0$, let $K_\lambda$ denote the family of all post-processing functions $\kappa : \mathbb{R} \to \mathbb{R}$ such that there exists a 1-Lipschitz function $\eta : \mathbb{R} \to [-1/\lambda, 1/\lambda]$ satisfying $\kappa(t) = t + \eta(t)$ for every $t \in \mathbb{R}$. Let $\psi$ and $\ell^{(\psi)}$ be defined as in Definition 4.3. Let $\mathcal{D}$ be a distribution over $\mathcal{X} \times \{0, 1\}$. We define the dual post-processing gap of a function $g : \mathcal{X} \to \mathbb{R}$ to be*

$$\mathsf{pGap}_{\mathcal{D}}^{(\psi,\lambda)}(g) := \mathbb{E}_{(x,y)\sim\mathcal{D}}\, \ell^{(\psi)}(y, g(x)) - \inf_{\kappa \in K_\lambda} \mathbb{E}_{(x,y)\sim\mathcal{D}}\, \ell^{(\psi)}\big(y, \kappa(g(x))\big).$$

Definition 4.4 is a generalization of Definition 2.2 from the squared loss to an arbitrary proper loss corresponding to a function $\psi$ as in (6). The following definition generalizes the definition of smooth calibration in Definition 2.3 to an arbitrary proper loss in a similar way. Here we use the fact proved in Appendix E (equation (20)) that if the function $\psi$ from Lemma 4.2 for a proper loss $\ell : \{0, 1\} \times V \to \mathbb{R}$ is differentiable, then $\nabla\psi(\mathsf{dual}(v)) = v$ holds for any $v \in V$, where $\nabla\psi(\cdot)$ denotes the derivative of $\psi$. This means that $\nabla\psi$ transforms a dual prediction to its original prediction.

**Definition 4.5** (Dual smooth calibration). *For $\lambda > 0$, let $H_\lambda$ be the family of all $\lambda$-Lipschitz functions $\eta : \mathbb{R} \to [-1, 1]$. Let $\psi : \mathbb{R} \to \mathbb{R}$ be a differentiable function with derivative $\nabla\psi(t) \in [0, 1]$ for every $t \in \mathbb{R}$. For a function $g : \mathcal{X} \to \mathbb{R}$, define predictor $f : \mathcal{X} \to [0, 1]$ such that $f(x) = \nabla\psi(g(x))$ for every $x \in \mathcal{X}$. Let $\mathcal{D}$ be a distribution over $\mathcal{X} \times \{0, 1\}$. We define the dual calibration error of $g$ to be*

$$\mathsf{smCE}_{\mathcal{D}}^{(\psi,\lambda)}(g) := \sup_{\eta \in H_\lambda} |\mathbb{E}_{(x,y)\sim\mathcal{D}}[(y - f(x))\eta(g(x))]|.$$

In Theorem 4.6 below we state our generalization of Theorem 2.4 to arbitrary proper loss functions. Theorem 4.6 shows that achieving a small dual post-processing gap is equivalent to achieving a small dual calibration error. We then show in Lemma 4.7 that a small dual calibration error implies a small (standard) smooth calibration error.

**Theorem 4.6.** *Let $\psi : \mathbb{R} \to \mathbb{R}$ be a differentiable convex function with derivative $\nabla\psi(t) \in [0, 1]$ for every $t \in \mathbb{R}$. For $\lambda > 0$, assume that $\psi$ is $\lambda$-smooth, i.e.,*

$$|\nabla\psi(t) - \nabla\psi(t')| \le \lambda|t - t'| \quad \text{for every } t, t' \in \mathbb{R}. \tag{8}$$

*Then for every $g : \mathcal{X} \to \mathbb{R}$ and any distribution $\mathcal{D}$ over $\mathcal{X} \times \{0, 1\}$,*

$$\mathsf{smCE}_{\mathcal{D}}^{(\psi,\lambda)}(g)^2/2 \le \lambda\, \mathsf{pGap}_{\mathcal{D}}^{(\psi,\lambda)}(g) \le \mathsf{smCE}_{\mathcal{D}}^{(\psi,\lambda)}(g).$$

**Lemma 4.7.** *Let $\psi : \mathbb{R} \to \mathbb{R}$ be a differentiable convex function with derivative $\nabla\psi(t) \in [0, 1]$ for every $t \in \mathbb{R}$. For $\lambda > 0$, assume that $\psi$ is $\lambda$-smooth as in (8). For $g : \mathcal{X} \to \mathbb{R}$, define $f : \mathcal{X} \to [0, 1]$ such that $f(x) = \nabla\psi(g(x))$ for every $x \in \mathcal{X}$. For a distribution $\mathcal{D}$ over $\mathcal{X} \times \{0, 1\}$, define $\mathsf{smCE}_{\mathcal{D}}(f)$ as in Definition 2.3. Then*

$$\mathsf{smCE}_{\mathcal{D}}(f) \le \mathsf{smCE}_{\mathcal{D}}^{(\psi,\lambda)}(g).$$

We defer the proofs of Theorem 4.6 and Lemma 4.7 to Appendix F.1 and Appendix F.2. Combining the two results, assuming $\psi$ is $\lambda$-smooth, we have

$$\mathsf{smCE}_{\mathcal{D}}(f)^2/2 \le \lambda\, \mathsf{pGap}_{\mathcal{D}}^{(\psi,\lambda)}(g).$$

This means that a small dual post-processing gap for a proper loss implies a small (standard) smooth calibration error. The cross-entropy loss $\ell$ is a proper loss where the corresponding $\psi$ from Lemma 4.2 is given by $\psi(t) = \ln(1 + e^t)$ and it is $1/4$-smooth. Setting $\lambda = 1/4$ in Theorem 4.6 and Lemma 4.7, we get (3) and (4) for the cross-entropy loss.

# 5 Optimization Algorithms and Implicit Regularization

As simple consequences of our results, there are several natural algorithms which explicitly minimize loss, but implicitly achieve good calibration. Here, we focus on one such algorithm: *structural risk minimization*, and discuss additional such algorithms in Appendix G. We also give informal intuitions connecting each algorithm to training DNNs in practice.

We show that structural risk minimization (SRM) on the population distribution is guaranteed to output well-calibrated predictors, under mild assumptions on the function family and regularizer. Recall, structural risk minimization considers a function family $\mathcal{F}$ equipped with some "complexity measure" $\mu : \mathcal{F} \to \mathbb{R}_{\ge 0}$. For example, we may take the family of bounded-width neural networks, with depth

as the complexity measure. SRM then minimizes a proper loss, plus a complexity-regularizer given by $\mu$:

$$f^* = \operatorname*{argmin}_{f \in \mathcal{F}} \mathbb{E}_{\mathcal{D}}[\ell(y, f(x)] + \lambda \mu(f).$$

The complexity measure $\mu$ is usually designed to control the capacity of the function family for generalization reasons, though we will not require such assumptions about $\mu$. Rather, we only require that $\mu(f)$ does not grow too quickly under composition with Lipshitz functions. That is, $\mu(\kappa \circ f)$ should be at most a constant greater than $\mu(f)$, for all Lipshitz functions $\kappa$. Now, as long as the function family $\mathcal{F}$ is also closed under composition, SRM is well-calibrated:

**Claim 5.1.** *Let $\mathcal{F}$ be a class of functions $f : \mathcal{X} \to [0,1]$ closed under composition with $K$, where we define $K$ as in Definition 2.2. That is, for $f \in \mathcal{F}, \kappa \in K$ we have $\kappa \circ f \in \mathcal{F}$. Let $\mu : \mathcal{F} \to \mathbb{R}_{\geq 0}$ be any complexity measure satisfying, for all $f \in \mathcal{F}, \kappa \in K : \quad \mu(\kappa \circ f) \leq \mu(f) + 1$. Then the minimizer $f^*$ of the regularized optimization problem*

$$f^* = \operatorname*{argmin}_{f \in \mathcal{F}} \mathbb{E}_{\mathcal{D}} \ell_{\mathsf{sq}}(y, f(x)) + \lambda \mu(f).$$

*satisfies* $\mathsf{pGap}_{\mathcal{D}}(f^*) \leq \lambda$, *and thus by Theorem 2.4 has small calibration error:* $\mathsf{smCE}_{\mathcal{D}}(f^*) \leq \sqrt{\lambda}$.

*Proof.* The proof is almost immediate. Let $\mathrm{MSE}_{\mathcal{D}}(f)$ denote $\mathbb{E}_{\mathcal{D}}[\ell_{\mathsf{sq}}(y, f(x))]$. Consider the solution $f^*$ and arbitrary $\kappa \in K$. Since $\kappa \circ f^* \in \mathcal{F}$, we have

$$\mathrm{MSE}_{\mathcal{D}}(f^*) + \lambda \mu(f^*) \leq \mathrm{MSE}_{\mathcal{D}}(\kappa \circ f^*) + \lambda \mu(\kappa \circ f^*) \leq \mathrm{MSE}_{\mathcal{D}}(\kappa \circ f^*) + \lambda \mu(f^*) + \lambda.$$

After rearranging, this is equivalent to

$$\mathrm{MSE}_{\mathcal{D}}(f^*) - \mathrm{MSE}_{\mathcal{D}}(\kappa \circ f^*) \leq \lambda,$$

and since $\kappa \in K$ was arbitrary, we get $\mathsf{pGap}_{\mathcal{D}}(f) \leq \lambda$. as desired. $\square$

**Discussion: Implicit Regularization.** This aspect of SRM connects our results to the "implicit regularization" hypothesis from deep learning theory community [Neyshabur, 2017]. The implicit regularization hypothesis is an informal belief that SGD on neural networks implicitly performs minimization on a "complexity-regularized" objective, which ensures generalization. The exact form of this complexity regularizer remains elusive, but many works have attempted to identify its structural properties (e.g. Arora et al. [2019], Neyshabur et al. [2014], Neyshabur [2017], Zhang et al. [2021]). In this context, our results imply that if the implicit regularization hypothesis is true, then as long as the complexity measure $\mu$ doesn't increase too much from composition, the final output of SGD will be well-calibrated.

## 6 Conclusion

Inspired by recent empirical observations, we studied formal conditions under which optimizing a proper loss also happens to yield calibration "for free." We identified a certain local optimality condition that characterizes distance to calibration, in terms of properties of the (proper) loss landscape. Our results apply even to realistic optimization algorithms, which optimize over restricted function families, and may not reach global minima within these families. In particular, our formal results suggest an intuitive explanation for why state-of-the-art DNNs are often well-calibrated: because their test loss cannot be improved much by adding a few more layers. It also offers guidance for how one can achieve calibration simply by proper loss minimization over a sufficiently expressive family of predictors.

**Limitations.** The connection between our theoretical results and DNNs in practice is heuristic— for example, we do not have a complete proof that training a DNN with SGD on natural distributions will yield a calibrated predictor, though we have formal results that give some intuition for this. Part of the difficulty here is due to known definitional barriers [Nakkiran, 2021], which is that we would need to formally specify what "DNN" means in practice (which architectures, and sizes?), what "SGD" means in practice (what initialization, learning-rate schedule, etc), and what "natural distributions" means in practice. Finally, we intuitively expect the results from our work to generalize beyond the binary label setting; we leave it to future work to formalize this intuition.

## Acknowledgments and Disclosure of Funding

Part of this work was performed while LH was interning at Apple. LH is also supported by Moses Charikar's and Omer Reingold's Simons Investigators awards, Omer Reingold's NSF Award IIS-1908774, and the Simons Foundation Collaboration on the Theory of Algorithmic Fairness. JB is supported by Simons Foundation.

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

# A  Additional Related Works

**Calibration without Bayesian Modeling.**    Our results show that it is formally possible to obtain calibrated classifiers via pure loss minimization, without incorporating Bayesian priors into the learning process. We view this as encouraging for the field of uncertainty quantification in machine learning, since it shows it is possible to obtain accurate uncertainties without incurring the computational and conceptual overheads of Bayesian methods [Abdar et al., 2021, Graves, 2011, MacKay, 1995, Gal, 2016].

**Calibration Methods.**    There have been many methods proposed to achieve accurate models that are also well-calibrated. The classic two-stage approach is to first learn an accurate classifier, and then post-process for calibration (via e.g. isotonic regression or Platt scaling as in Platt [1999], Zadrozny and Elkan [2001, 2002], Niculescu-Mizil and Caruana [2005]). Recent papers on neural networks also suggest optimizing for both calibration and accuracy jointly, by adding a "calibration regularizer" term to the standard accuracy-encouraging objective function [Kumar et al., 2018, Karandikar et al., 2021]. In contrast, we are interested in the setting where loss-minimization happens to yield calibration "out of the box", without re-calibration. There is a growing body of work on principled measures of calibration [Kakade and Foster, 2008, Gopalan et al., 2022b, Błasiok et al., 2023b], which we build on in our paper. Clarté et al. [2023a,b] analyze calibration for empirical risk minimization and highlight the role played by regularization.

**Multicalibration.**    There has recently been great interest in a generalization of calibration known as multicalibration [Hébert-Johnson et al., 2018], see also Kearns et al. [2017], motivated by algorithmic fairness considerations. A series of works explores the connection between multicalibration and minimization of convex losses, through the notion of omniprediction [Gopalan et al., 2022a, 2023a,b, Hu et al., 2022, Globus-Harris et al., 2023]. A tight characterization of multicalibration in terms of squared loss minimization is presented in Gopalan et al. [2023b], which involves a novel notion of loss minimization called swap agnostic learning. Another related result is the recent work of Błasiok et al. [2023a] which shows that minimizing squared loss for sufficiently deep neural nets yields multicalibration with regard to smaller neural nets of a fixed complexity. This result is similar in flavor since it argues that a lack of multicalibration points to a good way to reduce the loss, and relies on the expressive power of DNNs to do this improvement. But it goes only in one direction. Our results apply to any proper loss, and give a tight characterization of calibration error. We generalize our results to multicalibration in Appendix F.

# B  Miscalibration Example from Logistic Regression

Figure 2 gives an example of a 1-dimensional distribution for which logistic regression (on the population distribution itself) is severely mis-calibrated. This occurs despite logistic regression optimizing a proper loss. Note that the constant predictor $f(x) = 0.5$ is perfectly calibrated in this example, but logistic regression encourages the solution to *deviate* from it in order to minimize the proper cross-entropy loss. Similar examples of logistic regression being miscalibrated are found in Kull et al. [2017].

# C  Post-processing Gap for Cross Entropy Loss

Here we show that directly replacing the squared loss in the definition of post-processing gap (Definition 2.2) with the cross-entropy loss does not give us a similar result as Theorem 2.4. Specifically, we give an example where the smooth calibration error of a predictor is small but the post-processing gap w.r.t. the cross entropy loss can be arbitrarily large.

We choose $\mathcal{X} = \{x_0, x_1\}$ and let $\mathcal{D}$ be the distribution over $\mathcal{X} \times \{0, 1\}$ such that the marginal distribution over $\mathcal{X}$ is uniform, $\mathbb{E}_{(x,y)\sim\mathcal{D}}[y|x = x_0] = 0.1$ and $\mathbb{E}_{(x,y)\sim\mathcal{D}}[y|x = x_1] = 0.9$. Consider predictor $f : \mathcal{X} \to [0, 1]$ such that $f(x_0) = \varepsilon$ and $f(x_1) = 1 - \varepsilon$ for some $\varepsilon \in (0, 0.1)$. As $\varepsilon \to 0$, it is easy to verify that $\mathsf{smCE}_{\mathcal{D}}(f) \to 0.05$. However, the post-processing gap w.r.t. the cross-entropy loss tends to $+\infty$ because $\ell(1, \varepsilon) = \ell(0, 1 - \varepsilon) \to +\infty$ as $\varepsilon \to 0$ for the cross-entropy loss $\ell$.

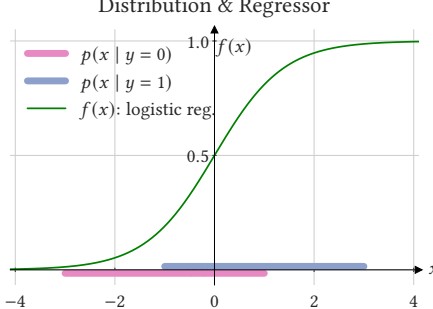
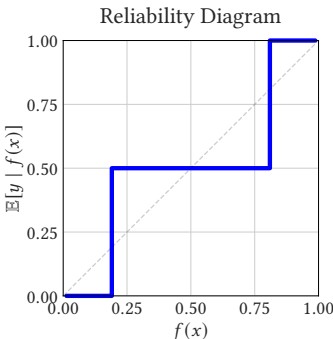

Figure 2: **Logistic Regression can be Miscalibrated.** We illustrate a 1-dimensional distribution on which logistic regression is severely miscalibrated: consider two overlapping uniform distributions, one for each class. Specifically, with probability $1/2$ we have $y = 0$ and $x \sim \mathrm{Unif}([-3, 1])$, and with the remaining probability $1/2$, we have $y = 1$ and $x \sim \mathrm{Unif}([-1, 3])$. This distribution is shown on the left, as well as the optimal logistic regressor $f$ on this distribution. In this setting, the logistic regressor is miscalibrated (as shown at right), despite logistic regression optimizing a proper loss. Note that the constant predictor $f(x) = 0.5$ is perfectly calibrated in this example, but logistic regression encourages the solution to *deviate* from it in order to minimize the proper cross-entropy loss. Similar examples of logistic regression being miscalibrated are also found in Kull et al. [2017].

## D   Tight Examples

We give examples showing that the constants in Theorem 2.4 and in Corollary 2.7 are all optimal. Specifically Lemma D.1 shows that the constants in the upper bound of pGap in Theorem 2.4 are optimal, whereas Lemma D.2 shows that the constants in the lower bound of pGap are also optimal. Lemma D.3 shows that the constants in the upper bound of $\mathsf{pGap}^{(\psi, 1/4)}$ in (3) are optimal. Lemma D.4 simultaneously shows that the constants in the lower bound of $\mathsf{pGap}^{(\psi, 1/4)}$ in (3) and the constants in (4) are optimal.

**Lemma D.1.** *Let $\mathcal{X} = \{x_0, x_1\}$ be a set of two individuals. For every $\varepsilon \in (0, 1/4)$, there exists a distribution $\mathcal{D}$ over $\mathcal{X} \times \{0, 1\}$ and a predictor $f : \mathcal{X} \to [0, 1]$ such that*

$$\mathsf{pGap}_{\mathcal{D}}(f) = \varepsilon - 3\varepsilon^2, \quad \text{whereas} \quad \mathsf{smCE}_{\mathcal{D}}(f) = \varepsilon/2 - \varepsilon^2.$$

*Proof.* We simply choose $\mathcal{D}$ to be the uniform distribution over $\{(x_0, 0), (x_1, 1)\}$, and we choose $f(x_0) = 1/2 - \varepsilon$ and $f(x_1) = 1/2 + \varepsilon$. For any 1-Lipschitz function $\eta : [0, 1] \to [-1, 1]$, we have

$$|\mathbb{E}_{(x,y)\sim\mathcal{D}}[(y - f(x))\eta(f(x))]| = |-(1/2)(1/2 - \varepsilon)\eta(f(x_0)) + (1/2)(1/2 - \varepsilon)\eta(f(x_1))|$$
$$= (1/2)(1/2 - \varepsilon)|\eta(1/2 - \varepsilon) - \eta(1/2 + \varepsilon)|.$$

Clearly, the supremum of the above quantity is $\varepsilon/2 - \varepsilon^2$ which is achieved when $|\eta(1/2 - \varepsilon) - \eta(1/2 + \varepsilon)| = 2\varepsilon$. Therefore,

$$\mathsf{smCE}_{\mathcal{D}}(f) = \varepsilon/2 - \varepsilon^2.$$

For any predictor $f' : \mathcal{X} \to [0, 1]$ and the squared loss $\ell$,

$$\mathbb{E}_{(x,y)\sim\mathcal{D}}[\ell(y, f'(x))] = (1/2)(0 - f'(x_0))^2 + (1/2)(1 - f'(x_1))^2$$
$$= f'(x_0)^2/2 + f'(x_1)^2/2 - f'(x_1) + 1/2$$
$$= \frac{1}{4}(f'(x_0) + f'(x_1) - 1)^2 + \frac{1}{4}(f'(x_1) - f'(x_0) - 1)^2$$

When $f'(x) = f(x) + \eta(f(x))$ for a 1-Lipschitz function $\eta : [0, 1] \to [-1, 1]$, the infimum of the above quantity is achieved when $f'(x_0) + f'(x_1) - 1 = 0$ and $f'(x_1) - f'(x_0) = 4\varepsilon$, and the infimum is $(1/4)(1 - 4\varepsilon)^2$. If we choose $f' = f$ instead, we get $\mathbb{E}[\ell(y, f(x))] = (1/4)(1 - 2\varepsilon)^2$. Therefore,

$$\mathsf{pGap}_{\mathcal{D}}(f) = (1/4)(1 - 2\varepsilon)^2 - (1/4)(1 - 4\varepsilon)^2 = \varepsilon - 3\varepsilon^2. \qquad \square$$

**Lemma D.2.** *Let $\mathcal{X} = \{x_0\}$ be a set consisting of only a single individual $x_0$. Let $\mathcal{D}$ be the uniform distribution over $\mathcal{X} \times \{0,1\}$ and for $\varepsilon \in (0,1/2)$, let $f : \mathcal{X} \to [0,1]$ be the predictor such that $f(x_0) = 1/2 + \varepsilon$. Then*

$$\mathsf{pGap}_{\mathcal{D}}(f) = \varepsilon^2, \quad \text{whereas} \quad \mathsf{smCE}_{\mathcal{D}}(f) = \varepsilon.$$

*Proof.* For any 1-Lipschitz function $\eta : [0,1] \to [-1,1]$,

$$|\mathbb{E}_{(x,y)\sim\mathcal{D}}[(y - f(x))\eta(f(x))]| = \varepsilon|\eta(1/2 + \varepsilon)|.$$

The supremum of the quantity above over $\eta$ is clearly $\varepsilon$, implying that $\mathsf{smCE}_{\mathcal{D}}(f) = \varepsilon$.

For any predictor $f' : \mathcal{X} \to [0,1]$ and the squared loss $\ell$,

$$\begin{aligned}
\mathbb{E}_{(x,y)\sim\mathcal{D}}[\ell(y, f'(x))] &= (1/2)(0 - f'(x_0))^2 + (1/2)(1 - f'(x_0))^2 \\
&= f'(x_0)^2 - f'(x_0) + 1/2 \\
&= 1/4 + (f'(x_0) - 1/2)^2.
\end{aligned}$$

The infimum of the above quantity over a function $f' : \mathcal{X} \to [0,1]$ satisfying $f'(x) = f(x) + \eta(f(x))$ for some 1-Lipschitz $\eta : [0,1] \to [-1,1]$ is clearly $1/4$. If we choose $f' = f$ instead, we get $\mathbb{E}_{(x,y)\sim\mathcal{D}}[\ell(y, f(x))] = 1/4 + \varepsilon^2$. Therefore,

$$\mathsf{pGap}_{\mathcal{D}}(f) = (1/4 + \varepsilon^2) - 1/4 = \varepsilon^2. \qquad \square$$

**Lemma D.3.** *Define $\mathsf{pGap}^{(\psi,1/4)}$ and $\mathsf{smCE}^{(\psi,1/4)}$ as in Definition 2.5 and Definition 2.6 for the cross-entropy loss. Let $\mathcal{X} = \{x_0, x_1\}$ be a set of two individuals. For every $\varepsilon \in (0, 1/4)$, there exists a distribution $\mathcal{D}$ over $\mathcal{X} \times \{0,1\}$ and a function $g : \mathcal{X} \to \mathbb{R}$ such that*

$$\mathsf{pGap}_{\mathcal{D}}^{(\psi,1/4)}(g) \geq 2\varepsilon + O(\varepsilon^2), \quad \text{whereas} \quad \mathsf{smCE}_{\mathcal{D}}^{(\psi,1/4)}(g) \leq \varepsilon/2 + O(\varepsilon^2).$$

*Proof.* We simply choose $\mathcal{D}$ to be the uniform distribution over $\{(x_0, 0), (x_1, 1)\}$, and we choose $g(x_0) = -4\varepsilon$ and $g(x_1) = 4\varepsilon$. Let $\sigma$ denote the sigmoid transformation given by $\sigma(t) = e^t/(1 + e^t)$. The Taylor expansion of the sigmoid transformation $\sigma$ around $t = 0$ is $\sigma(t) = 1/2 + O(t)$. Thus the predictor $f$ we get from applying $\sigma$ to $g$ satisfies $f(x_1) = 1 - f(x_0) = 1/2 + O(\varepsilon)$. For any $1/4$-Lipschitz function $\eta : \mathbb{R} \to [-1,1]$, we have

$$\begin{aligned}
\mathbb{E}_{(x,y)\sim\mathcal{D}}[(y - f(x))\eta(g(x))] &= (1/2)(0 - f(x_0))\eta(g(x_0)) + (1/2)(1 - f(x_1))\eta(g(x_1)) \\
&= -(1/2)(1 - f(x_1))\eta(g(x_0)) + (1/2)(1 - f(x_1))\eta(g(x_1)) \\
&= (1/2)(1 - f(x_1))(\eta(g(x_1)) - \eta(g(x_0))) \\
&\leq (1/2)(1 - f(x_1))|g(x_0) - g(x_1)|/4 \\
&\leq (1/2)(1/2 + O(\varepsilon)) \cdot (8\varepsilon)/4 \\
&= \varepsilon/2 + O(\varepsilon^2).
\end{aligned}$$

This implies that $\mathsf{smCE}_{\mathcal{D}}^{(\psi,1/4)}(g) \leq \varepsilon/2 + O(\varepsilon^2)$.

For any function $g' : \mathcal{X} \to \mathbb{R}$ and the logistic loss $\ell^{(\psi)}$,

$$\begin{aligned}
\mathbb{E}_{(x,y)\sim\mathcal{D}}[\ell^{(\psi)}(y, g'(x))] &= (1/2)\ln(1 + e^{g'(x_0)}) + (1/2)\ln(1 + e^{g'(x_1)}) - (1/2)g'(x_1) \\
&= (1/2)\ln(1 + e^{g'(x_0)}) + (1/2)\ln(1 + e^{-g'(x_1)}).
\end{aligned}$$

We use the Taylor expansion $\ln(1 + e^t) = \ln 2 + t/2 + O(t^2)$ to get

$$\mathbb{E}_{(x,y)\sim\mathcal{D}}[\ell^{(\psi)}(y, g'(x))] = \ln 2 + (1/4)(g'(x_0) - g'(x_1)) + O(g'(x_0)^2 + g'(x_1)^2). \tag{9}$$

Consider the case when $g'(x_0) = -8\varepsilon$ and $g'(x_1) = 8\varepsilon$. Clearly, $g'$ can be written as $g'(x) = g(x) + \eta(g(x))$ for a 1-Lipschitz function $\eta : \mathbb{R} \to [-4, 4]$. By (9), the expected loss achieved by $g'$ is

$$\mathbb{E}_{(x,y)\sim\mathcal{D}}[\ell^{(\psi)}(y, g'(x))] = \ln 2 - 4\varepsilon + O(\varepsilon^2).$$

If $g' = g$ instead, the expected loss is $\ln 2 - 2\varepsilon + O(\varepsilon^2)$. Taking the difference between the two, we get

$$\mathsf{pGap}_{\mathcal{D}}^{(\psi,1/4)}(g) \geq 2\varepsilon + O(\varepsilon^2). \qquad \square$$

**Lemma D.4.** *Define* $\mathsf{pGap}^{(\psi, 1/4)}$ *as in Definition 2.5 for the cross-entropy loss. Let* $\mathcal{X} = \{x_0\}$ *be a set consisting of only a single individual* $x_0$. *Let* $\mathcal{D}$ *be the uniform distribution over* $\mathcal{X} \times \{0, 1\}$ *and for* $\varepsilon \in (0, 1/2)$, *let* $g : \mathcal{X} \to \mathbb{R}$ *be the function such that* $g(x_0) = 4\varepsilon$. *Then*

$$\mathsf{pGap}_{\mathcal{D}}^{(\psi, 1/4)}(g) = 2\varepsilon^2 + O(\varepsilon^4), \quad \text{whereas} \quad \mathsf{smCE}_{\mathcal{D}}(f) = \varepsilon + O(\varepsilon^3),$$

*where* $f : \mathcal{X} \to (0, 1)$ *is given by* $f(x) = \sigma(g(x))$ *for the sigmoid transformation* $\sigma(t) = e^t/(1 + e^t)$.

*Proof.* The Taylor expansion of $\sigma$ around $t = 0$ is $\sigma(t) = 1/2 + t/4 + O(t^3)$. Thus $f(x_0) = \sigma(g(x_0)) = 1/2 + \varepsilon + O(\varepsilon^3)$. For any 1-Lipschitz function $\eta : [0, 1] \to [-1, 1]$,

$$|\mathbb{E}_{(x,y) \sim \mathcal{D}}[(y - f(x)\eta(f(x))]| = |(1/2 - f(x_0))\eta(f(x_0))| = (\varepsilon + O(\varepsilon^3))|\eta(f(x_0))|.$$

The supremum of the quantity above over $\eta$ is clearly $\varepsilon + O(\varepsilon^3)$, implying that $\mathsf{smCE}_{\mathcal{D}}(f) = \varepsilon + O(\varepsilon^3)$.

For any function $g' : \mathcal{X} \to [0, 1]$ and the logistic loss $\ell^{(\psi)}$,

$$\mathbb{E}_{(x,y) \sim \mathcal{D}}[\ell^{(\psi)}(y, g'(x))] = \ln(1 + e^{g'(x_0)}) - (1/2)g'(x_0) \tag{10}$$
$$= \ln(e^{g'(x_0)/2} + e^{-g'(x_0)/2}).$$

The infimum of the above quantity over a function $g' : \mathcal{X} \to \mathbb{R}$ satisfying $g'(x) = g(x) + \eta(g(x))$ for some 1-Lipschitz $\eta : \mathbb{R} \to [-4, 4]$ is $\ln 2$ achieved when $g'(x_0) = 0$. If we choose $g' = g$ instead, we get $\mathbb{E}_{(x,y) \sim \mathcal{D}}[\ell^{(\psi)}(y, g(x))] = \ln 2 + 2\varepsilon^2 + O(\varepsilon^4)$ by plugging the Taylor expansion $\ln(1 + e^t) = \ln 2 + t/2 + t^2/8 + O(t^4)$ into (10). Therefore,

$$\mathsf{pGap}_{\mathcal{D}}^{(\psi, 1/4)}(g) = 2\varepsilon^2 + O(\varepsilon^4). \qquad \square$$

# E Proper Loss and Convex Functions

It is known that there is a correspondence between proper loss functions and convex functions [Shuford et al., 1966, Savage, 1971, Schervish, 1989, Buja et al., 2005]. Here we demonstrate this correspondence, which is important for extending the connection between smooth calibration and post-processing gap (Theorem 2.4) to general proper loss functions in Section 4.

**Definition E.1** (Sub-gradient). *For a function* $\varphi : V \to \mathbb{R}$ *defined on* $V \subseteq \mathbb{R}$, *we say* $t \in \mathbb{R}$ *is a sub-gradient of* $\varphi$ *at* $v \in V$ *if*

$$\varphi(v') \geq \varphi(v) + (v' - v)t \quad \text{for every } v' \in V,$$

*or equivalently,*

$$vt - \varphi(v) = \max_{v' \in V} (v't - \varphi(v')). \tag{11}$$

**Definition E.2** (Fenchel-Young divergence). *For a pair of functions* $\varphi : V \to \mathbb{R}$ *and* $\psi : T \to \mathbb{R}$ *defined on* $V, T \subseteq \mathbb{R}$, *we define the* Fenchel-Young divergence $D_{\varphi, \psi} : V \times T \to \mathbb{R}$ *such that*

$$D_{\varphi, \psi}(v, t) = \varphi(v) + \psi(t) - vt \quad \text{for every } v \in V \text{ and } t \in T.$$

The following claim follows immediately from the two definitions above.

**Claim E.3.** *Let* $\varphi : V \to \mathbb{R}$ *and* $\psi : T \to \mathbb{R}$ *be functions defined on* $V, T \subseteq \mathbb{R}$. *Then* $t \in T$ *is a sub-gradient of* $\varphi$ *at* $v \in V$ *if and only if* $D_{\varphi, \psi}(v, t) = \min_{v' \in V} D_{\varphi, \psi}(v', t)$. *Similarly,* $v$ *is a sub-gradient of* $\psi$ *at* $t$ *if and only if* $D_{\varphi, \psi}(v, t) = \min_{t' \in T} D_{\varphi, \psi}(v, t')$. *In particular, assuming* $D_{\varphi, \psi}(v', t') \geq 0$ *for every* $v' \in V$ *and* $t' \in T$ *whereas* $D_{\varphi, \psi}(v, t) = 0$ *for some* $v \in V$ *and* $t \in T$, *then* $t$ *is a sub-gradient of* $\varphi$ *at* $v$, *and* $v$ *is a sub-gradient of* $\psi$ *at* $t$.

**Lemma E.4** (Convex functions from proper loss). *Let* $V \subseteq [0, 1]$ *be a non-empty interval. Let* $\ell : \{0, 1\} \times V \to \mathbb{R}$ *be a proper loss function. For every* $v \in V$, *define* $\mathsf{dual}(v) := \ell(0, v) - \ell(1, v)$. *There exist convex functions* $\varphi : V \to \mathbb{R}$ *and* $\psi : \mathbb{R} \to \mathbb{R}$ *such that*

$$\ell(y, v) = \psi(\mathsf{dual}(v)) - y\,\mathsf{dual}(v) \qquad \text{for every } y \in \{0, 1\} \text{ and } v \in V, \tag{12}$$
$$D_{\varphi, \psi}(v, t) \geq 0 \qquad \text{for every } v \in V \text{ and } t \in \mathbb{R}, \tag{13}$$
$$D_{\varphi, \psi}(v, \mathsf{dual}(v)) = 0 \qquad \text{for every } v \in V, \tag{14}$$
$$0 \leq \frac{\psi(t_2) - \psi(t_1)}{t_2 - t_1} \leq 1 \qquad \text{for every distinct } t_1, t_2 \in \mathbb{R}. \tag{15}$$

Before we prove the lemma, we remark that (12) and (14) together imply the following:

$$\ell(y, v) = -\varphi(v) + (v - y)\mathsf{dual}(v) \quad \text{for every } y \in \{0, 1\} \text{ and } v \in V.$$

*Proof.* Any loss function $\ell : \{0, 1\} \times V \to \mathbb{R}$, proper or not, can be written as $\ell(y, v) = \ell(0, v) - y\,\mathsf{dual}(v)$. When $\ell$ is proper, it is easy to see that for $v, v' \in V$ with $\mathsf{dual}(v) = \mathsf{dual}(v')$, it holds that $\ell(0, v) = \ell(0, v')$; otherwise, $\ell(y, v)$ is always smaller or always larger than $\ell(y, v')$ for every $y \in \{0, 1\}$, making it impossible for $\ell$ to be proper. Therefore, for proper $\ell$, there exists $\psi_0 : T \to \mathbb{R}$ such that $\ell(0, v) = \psi_0(\mathsf{dual}(v))$ for every $v \in V$, where $T := \{\mathsf{dual}(v) : v \in V\}$. Therefore,

$$\ell(y, v) = \ell(0, v) - y\,\mathsf{dual}(v) = \psi_0(\mathsf{dual}(v)) - y\,\mathsf{dual}(v). \tag{16}$$

By the assumption that $\ell$ is proper, for every $v \in V$,

$$v \in \operatorname*{argmin}_{v' \in V} \ \mathbb{E}_{y \sim \mathsf{Ber}(v)} \ \ell(y, v') = \operatorname*{argmin}_{v' \in V} \Big( \psi_0(\mathsf{dual}(v')) - v\,\mathsf{dual}(v') \Big),$$

and thus

$$\mathsf{dual}(v) \in \operatorname*{argmin}_{t \in T} \Big( \psi_0(t) - vt \Big).$$

For every $v \in V$, we define

$$\varphi(v) := v\,\mathsf{dual}(v) - \psi_0(\mathsf{dual}(v)) = \max_{t \in T} \Big( vt - \psi_0(t) \Big).$$

The definition of $\varphi$ immediately implies the following:

$$D_{\varphi, \psi_0}(v, t) \geq 0 \qquad \qquad \text{for every } v \in V \text{ and } t \in T, \tag{17}$$
$$D_{\varphi, \psi_0}(v, \mathsf{dual}(v)) = 0 \qquad \qquad \text{for every } v \in V. \tag{18}$$

By Claim E.3, $\mathsf{dual}(v)$ is a sub-gradient of $\varphi$ at $v$. This proves that $\varphi$ is convex. Now we define $\psi : \mathbb{R} \to \mathbb{R}$ such that $\psi(t) = \sup_{v \in V} vt - \varphi(v)$ for every $t \in \mathbb{R}$. This definition ensures that inequality (13) holds. By Lemma H.3, $\psi$ is a convex function and (15) holds. Moreover, for every $v \in V$, we have shown that $\mathsf{dual}(v)$ is a sub-gradient of $\varphi$ at $v$, so by (11), $\psi(\mathsf{dual}(v)) = v\,\mathsf{dual}(v) - \varphi(v) = \psi_0(\mathsf{dual}(v))$. This implies that (12) and (14) hold because of (16) and (18). $\qquad \square$

**Lemma E.5** (Proper loss from convex functions). *Let $V \subseteq [0, 1]$ be a non-empty interval. Let $\varphi : V \to \mathbb{R}$, $\psi : \mathbb{R} \to \mathbb{R}$, and $\mathsf{dual} : V \to \mathbb{R}$ be functions satisfying (13) and (14). Define loss function $\ell : \{0, 1\} \times V \to \mathbb{R}$ as in (12). Then*

1. *$\ell$ is a proper loss function;*

2. *$\varphi$ is convex;*

3. *for every $v \in V$, $\mathsf{dual}(v)$ is a sub-gradient of $\varphi$ at $v$, and $v$ is a sub-gradient of $\psi$ at $\mathsf{dual}(v)$;*

4. *$\varphi(v) = v\,\mathsf{dual}(v) - \psi(\mathsf{dual}(v)) = -\mathbb{E}_{y \sim \mathsf{Ber}(v)}[\ell(y, v)]$;*

5. *if we define $\psi'(t) := \sup_{v \in V}(vt - \varphi(v))$, then $\psi'$ is convex and $\psi'(\mathsf{dual}(v)) = \psi(\mathsf{dual}(v))$ for every $v \in V$. Moreover, for any two distinct real numbers $t_1, t_2$,*

$$0 \leq \frac{\psi'(t_2) - \psi'(t_1)}{t_2 - t_1} \leq 1. \tag{19}$$

*Proof.* For every $v \in V$, equations (13) and (14) imply that

$$\mathsf{dual}(v) \in \operatorname*{argmin}_{t \in \mathbb{R}} \big( \psi(t) - vt \big),$$

and thus

$$v \in \operatorname*{argmin}_{v' \in V} \Big( \psi(\mathsf{dual}(v')) - v\,\mathsf{dual}(v') \Big) = \operatorname*{argmin}_{v' \in V} \ \mathbb{E}_{y \sim \mathsf{Ber}(v)} \ \ell(y, v').$$

This proves that $\ell$ is proper (Item 1). Item 3 follows from (13), (14) and Claim E.3. Item 2 follows from Item 3. The first equation in Item 4 follows from (14), and the second equation follows from (12). For Item 5, the convexity of $\psi'$ and inequality (19) follow from Lemma H.3, and $\psi'(\mathsf{dual}(v)) = \psi(\mathsf{dual}(v))$ holds because $\psi'(\mathsf{dual}(v)) = v\,\mathsf{dual}(v) - \varphi(v)$ by Item 3 and (11), and $\psi(\mathsf{dual}(v)) = v\,\mathsf{dual}(v) - \varphi(v)$ by Item 4. $\qquad \square$

Our discussion above gives a correspondence between a proper loss function $\ell : \{0,1\} \times V \to \mathbb{R}$ and a tuple $(\varphi, \psi, \mathsf{dual})$ satisfying (13) and (14). By Lemma E.5, if $\varphi, \psi$ and dual satisfy (13) and (14), then every $v \in V$ is a sub-gradient of $\psi$ at $\mathsf{dual}(v)$. Therefore, assuming $\psi$ is differentiable and using $\nabla\psi : \mathbb{R} \to \mathbb{R}$ to denote its derivative, if $t = \mathsf{dual}(v)$ for some $v \in V$, then we can compute $v$ from $t$ by

$$v = \nabla\psi(t). \tag{20}$$

For the cross entropy loss $\ell(y,v) = -y\ln v - (1-y)\ln(1-v)$, we can find the corresponding $\varphi, \psi, \mathsf{dual}$ and $\ell^{(\psi)}$ (see Definition 4.3) as follows:

$$\varphi(v) = - \mathop{\mathbb{E}}_{y\sim\mathsf{Ber}(v)}[\ell(y,v)] = v\ln v + (1-v)\ln(1-v),$$

$$\mathsf{dual}(v) = \ell(0,v) - \ell(1,v) = \ln(v/(1-v)),$$

$$\psi(t) = \sup_{v\in(0,1)}\big(vt - \varphi(v)\big) = \ln(1+e^t),$$

$$\ell^{(\psi)}(y,t) = \psi(t) - yt = \ln(1+e^t) - yt, \qquad\qquad \text{(logistic loss)}$$

$$\nabla\psi(t) = e^t/(1+e^t). \qquad\qquad \text{(sigmoid transformation)}$$

For the squared loss $\ell(y,v) = (y-v)^2$, we can find the corresponding $\varphi, \psi, \mathsf{dual}$ and $\ell^{(\psi)}$ as follows:

$$\varphi(v) = - \mathop{\mathbb{E}}_{y\sim\mathsf{Ber}(v)}[\ell(y,v)] = -v(1-v)^2 - (1-v)v^2 = v(v-1),$$

$$\mathsf{dual}(v) = \ell(0,v) - \ell(1,v) = 2v - 1,$$

$$\psi(t) = \sup_{v\in[0,1]}\big(vt - \varphi(v)\big) = \begin{cases} 0, & \text{if } t < -1; \\ (t+1)^2/4, & \text{if } -1 \le t \le 1; \\ t, & \text{if } t > 1. \end{cases}$$

$$\ell^{(\psi)}(y,t) = \psi(t) - yt = \begin{cases} -yt, & \text{if } t < -1; \\ (y - (t+1)/2)^2, & \text{if } -1 \le t \le 1; \\ (1-y)t, & \text{if } t > 1. \end{cases}$$

$$\nabla\psi(t) = \begin{cases} 0, & \text{if } t < -1; \\ (t+1)/2, & \text{if } -1 \le t \le 1; \\ 1, & \text{if } t > 1. \end{cases}$$

For the squared loss, one can alternatively choose $\psi(t) = (t+1)^2/4$ for *every* $t \in \mathbb{R}$ and accordingly define $\ell^{(\psi)}(y,t) = (y-(t+1)/2)^2$ for every $y \in \{0,1\}$ and $t \in \mathbb{R}$. This still ensures (12), (13) and (14), but (15) no longer holds.

## F  Generalized Dual (Multi)calibration and Generalized Dual Post-processing Gap

Below we generalize the notions of dual post-processing gap and dual smooth calibration error from Section 4 by replacing the Lipschitz functions in those definitions with functions from a general class. Here we allow the functions to not only depend on the dual prediction $g(x)$, but also depend directly on $x$. With such generality, the calibration notion we consider in Definition F.2 below captures the notion of multicalibration [Hébert-Johnson et al., 2018]. We connect this generalized calibration notion with a generalized notion of post-processing gap (Definition F.1) in Theorem F.3.

**Definition F.1** (Generalized dual post-processing gap). *Let $\psi$ and $\ell^{(\psi)}$ be defined as in Definition 4.3. Let $W$ be a class of functions $w : \mathcal{X} \times \mathbb{R} \to \mathbb{R}$. Let $\mathcal{D}$ be a distribution over $\mathcal{X} \times \{0,1\}$. We define the* generalized dual post-processing gap *of a function $g : \mathcal{X} \to \mathbb{R}$ w.r.t. class $W$ and distribution $\mathcal{D}$ to be*

$$\mathsf{genGap}_{\mathcal{D}}^{(\psi,W)}(g) := \mathbb{E}_{(x,y)\sim\mathcal{D}}\, \ell^{(\psi)}(y,g(x)) - \inf_{w\in W} \mathbb{E}_{(x,y)\sim\mathcal{D}}\, \ell^{(\psi)}\big(y, w(x,g(x))\big).$$

**Definition F.2** (Generalized dual calibration). *Let $\psi : \mathbb{R} \to \mathbb{R}$ be a differentiable function. For a function $g : \mathcal{X} \to \mathbb{R}$, define predictor $f : \mathcal{X} \to \mathbb{R}$ such that $f(x) = \nabla\psi(g(x))$ for every $x \in \mathcal{X}$.[4] Let*

---

[4]It is possible here for $f(x)$ to lie outside the interval $[0,1]$, in which case $f$ is not a valid predictor. However, if $\psi$ satisfies (15), then $\nabla\psi(t) \in [0,1]$ for every $t \in \mathbb{R}$ and thus $f(x) \in [0,1]$ for every $x \in \mathcal{X}$. Also, one can

$W$ be a class of functions $w : \mathcal{X} \times \mathbb{R} \to \mathbb{R}$. Let $\mathcal{D}$ be a distribution over $\mathcal{X} \times \{0, 1\}$. We define the generalized dual calibration error *of $g$ w.r.t. class $W$ and distribution $\mathcal{D}$ to be*

$$\mathsf{genCE}_{\mathcal{D}}^{(\psi, W)}(g) := \sup_{w \in W} |\mathbb{E}_{(x,y) \sim \mathcal{D}}[(y - f(x))w(x, g(x))]|.$$

**Theorem F.3.** *Let $\psi : \mathbb{R} \to \mathbb{R}$ be a differentiable convex function. For $\lambda > 0$, assume that $\psi$ is $\lambda$-smooth as defined in (8). Let $W$ be a class of bounded functions $w : \mathcal{X} \times \mathbb{R} \to [-1, 1]$. Define $W'$ to be the class of functions $w' : \mathcal{X} \times \mathbb{R} \to \mathbb{R}$ such that there exist $w \in W$ and $\beta \in [-1/\lambda, 1/\lambda]$ satisfying*

$$w'(x, t) = t + \beta w(x, t) \quad \text{for every } x \in \mathcal{X} \text{ and } t \in \mathbb{R}.$$

*Then for every $g : \mathcal{X} \to \mathbb{R}$ and any distribution $\mathcal{D}$ over $\mathcal{X} \times \{0, 1\}$,*

$$\mathsf{genCE}_{\mathcal{D}}^{(\psi, W)}(g)^2 / 2 \le \lambda \, \mathsf{genGap}_{\mathcal{D}}^{(\psi, W')}(g) \le \mathsf{genCE}_{\mathcal{D}}^{(\psi, W)}(g).$$

*Proof.* By the definition of $\ell^{(\psi)}$ in Definition 4.3,

$$\underset{(x,y) \sim \mathcal{D}}{\mathbb{E}} \ell^{(\psi)}(y, g(x)) - \underset{(x,y) \sim \mathcal{D}}{\mathbb{E}} \ell^{(\psi)}\big(y, w'(x, g(x))\big)$$

$$= \mathbb{E}[\psi(g(x)) - yg(x) - \psi(w'(x, g(x))) + yw'(x, g(x))]$$

$$= \mathbb{E}[\psi(g(x)) - \psi(w'(x, g(x))) + y\beta w(x, g(x))]. \tag{21}$$

By the convexity of $\psi$ and the $\lambda$-smoothness property, for every $t, t' \in \mathbb{R}$,

$$\nabla \psi(t)(t' - t) \le \psi(t') - \psi(t) = \int_t^{t'} \nabla \psi(\tau) \mathrm{d}\tau$$

$$\le \int_t^{t'} (\nabla \psi(t) + \lambda(\tau - t)) \mathrm{d}\tau = \nabla \psi(t)(t' - t) + \frac{\lambda}{2}(t' - t)^2. \tag{22}$$

For $x \in X$, setting $t$ to be $g(x)$ and setting $t'$ to be $w'(x, g(x)) := g(x) + \beta w(x, g(x))$ for some $w \in W$ and $\beta \in [-1/\lambda, 1/\lambda]$, we have

$$t' - t = w'(x, g(x)) - g(x) = \beta w(x, g(x)).$$

Plugging this into (22) and defining $f(x) := \nabla \psi(g(x)) = \nabla \psi(t)$, we have

$$f(x)\beta w(x, g(x)) \le \psi(w'(x, g(x))) - \psi(g(x)) \le f(x)\beta w(x, g(x)) + \frac{\lambda \beta^2}{2} w(x, g(x))^2.$$

Plugging this into (21) and using the fact that $|w(x, g(x))| \le 1$, we have

$$\mathbb{E}[(y - f(x))\beta w(x, g(x))] - \frac{\lambda \beta^2}{2}$$

$$\le \underset{(x,y) \sim \mathcal{D}}{\mathbb{E}} \ell^{(\psi)}(y, g(x)) - \underset{(x,y) \sim \mathcal{D}}{\mathbb{E}} \ell^{(\psi)}\big(y, w'(x, g(x))\big) \tag{23}$$

$$\le \mathbb{E}[(y - f(x))\beta w(x, g(x))]. \tag{24}$$

For any $w \in W$, choosing $\beta = \mathbb{E}[(y - f(x))w(x, g(x))]/\lambda$ and choosing $w' \in W'$ such that $w'(x, t) = t + \beta w(x, t)$, by (23) we have

$$\frac{1}{2\lambda} \mathbb{E}[(y - f(x))w(x, g(x))]^2 \le \underset{(x,y) \sim \mathcal{D}}{\mathbb{E}} \ell^{(\psi)}(y, g(x)) - \underset{(x,y) \sim \mathcal{D}}{\mathbb{E}} \ell^{(\psi)}\big(y, w'(x, g(x))\big).$$

This proves that

$$\mathsf{genCE}_{\mathcal{D}}^{(\psi, W)}(g)^2 / 2 \le \lambda \, \mathsf{genGap}_{\mathcal{D}}^{(\psi, W')}(g).$$

For any $w' \in W'$, there exists $\beta \in [-1/\lambda, 1/\lambda]$ and $w \in W$ such that $w'(x, t) = t + \beta w(x, t)$. By (24),

$$\underset{(x,y) \sim \mathcal{D}}{\mathbb{E}} \ell^{(\psi)}(y, g(x)) - \underset{(x,y) \sim \mathcal{D}}{\mathbb{E}} \ell^{(\psi)}\big(y, w'(x, g(x))\big)$$

$$\le \mathbb{E}[(y - f(x))\beta w(x, g(x))]$$

$$\le (1/\lambda) |\mathbb{E}[(y - f(x))w(x, g(x))]|.$$

This proves that

$$\lambda \, \mathsf{genGap}_{\mathcal{D}}^{(W')}(g) \le \mathsf{genCE}_{\mathcal{D}}^{(W)}(g). \qquad \square$$

start from a valid predictor $f : \mathcal{X} \to [0, 1]$ and define $g(x) = \mathsf{dual}(f(x))$. By Lemma E.5, this ensures that $f(x) = \nabla \psi(g(x))$ assuming that $(\varphi, \psi, \mathsf{dual})$ satisfy (13) and (14).

### F.1 Proof of Theorem 4.6

We restate and prove Theorem 4.6.

**Theorem 4.6.** *Let $\psi : \mathbb{R} \to \mathbb{R}$ be a differentiable convex function with derivative $\nabla\psi(t) \in [0,1]$ for every $t \in \mathbb{R}$. For $\lambda > 0$, assume that $\psi$ is $\lambda$-smooth, i.e.,*

$$|\nabla\psi(t) - \nabla\psi(t')| \le \lambda|t - t'| \quad \text{for every } t, t' \in \mathbb{R}. \tag{8}$$

*Then for every $g : \mathcal{X} \to \mathbb{R}$ and any distribution $\mathcal{D}$ over $\mathcal{X} \times \{0, 1\}$,*

$$\mathsf{smCE}_{\mathcal{D}}^{(\psi,\lambda)}(g)^2/2 \le \lambda\,\mathsf{pGap}_{\mathcal{D}}^{(\psi,\lambda)}(g) \le \mathsf{smCE}_{\mathcal{D}}^{(\psi,\lambda)}(g).$$

*Proof.* Let $W$ be the family of all $\lambda$-Lipschitz functions $\eta : \mathbb{R} \to [-1, 1]$ and define $W'$ as in Theorem F.3. We have

$$\mathsf{genCE}_{\mathcal{D}}^{(\psi,W)}(g) = \mathsf{smCE}_{\mathcal{D}}^{(\psi,\lambda)}(g), \quad \text{and}$$
$$\mathsf{genGap}_{\mathcal{D}}^{(\psi,W')}(g) = \mathsf{pGap}_{\mathcal{D}}^{(\psi,\lambda)}(g).$$

Theorem 4.6 then follows immediately from Theorem F.3. $\qquad\square$

### F.2 Proof of Lemma 4.7

We restate and prove Lemma 4.7.

**Lemma 4.7.** *Let $\psi : \mathbb{R} \to \mathbb{R}$ be a differentiable convex function with derivative $\nabla\psi(t) \in [0, 1]$ for every $t \in \mathbb{R}$. For $\lambda > 0$, assume that $\psi$ is $\lambda$-smooth as in (8). For $g : \mathcal{X} \to \mathbb{R}$, define $f : \mathcal{X} \to [0, 1]$ such that $f(x) = \nabla\psi(g(x))$ for every $x \in \mathcal{X}$. For a distribution $\mathcal{D}$ over $\mathcal{X} \times \{0,1\}$, define $\mathsf{smCE}_{\mathcal{D}}(f)$ as in Definition 2.3. Then*

$$\mathsf{smCE}_{\mathcal{D}}(f) \le \mathsf{smCE}_{\mathcal{D}}^{(\psi,\lambda)}(g).$$

*Proof.* For a 1-Lipschitz function $\eta : [0, 1] \to [-1, 1]$, define $\eta' : \mathbb{R} \to [-1, 1]$ such that $\eta'(t) = \eta(\nabla\psi(t))$ for every $t \in \mathbb{R}$. For $t_1, t_2 \in \mathbb{R}$, by the $\lambda$-smoothness property of $\eta$,

$$|\eta'(t_1) - \eta'(t_2)| = |\eta(\nabla\psi(t_1)) - \eta(\nabla\psi(t_2))| \le |\nabla\psi(t_1) - \nabla\psi(t_2)| \le \lambda|t_1 - t_2|.$$

This proves that $\eta'$ is $\lambda$-Lipschitz. Therefore,

$$|\,\mathbb{E}_{(x,y)\sim\mathcal{D}}[(y - f(x))\eta(f(x))]| = |\,\mathbb{E}_{(x,y)\sim\mathcal{D}}(y - f(x))\eta'(g(x))| \le \mathsf{smCE}_{\mathcal{D}}^{(\psi,\lambda)}(g).$$

Taking supremum over $\eta$ completes the proof. $\qquad\square$

## G   Optimization Algorithms Which Achieve Calibration

Continuing the discussion of structural risk minimization in Section 5, here we describe additional families of learning algorithms which explicitly minimize loss, but implicitly achieve good calibration.

### G.1   Iterative Risk Minimization

One intuition for why state-of-the-art DNNs are often calibrated is that they are "locally optimal" (in our sense) by design. For a state-of-the-art network, if it were possible to improve its test loss significantly by just adding a layer (and training it optimally), then the human practitioner training the DNN would have added another layer and re-trained. Thus, we expect the network eventually produced will be approximately loss-optimal with respect to adding a layer, and thus also with respect to univariate Lipshitz post-processings (so $\mathsf{pGap}_{\mathcal{D}}(f) \approx 0$).

This intuition makes several strong assumptions, for example, the practitioner must be able to re-train the last two layers globally optimally[5]. However, the same idea can apply in settings where optimization is tractable. We give one way of formalizing this via the Iterative Risk Minimization algorithm.

We work with the following setup:

---

[5]Technically, we only require a weaker statement of the form: For all networks $f_d$ of depth $d$ in the support of SGD outputs, if the loss of $\kappa \circ f_d$ is $\varepsilon$, then training a network $f_{d+1}$ of depth $d + 1$ will reach loss at most $\varepsilon$.

- We are interested in models $f \in \mathcal{F}$ with a complexity measure $\mu : \mathcal{F} \to \mathbb{Z}$. Think of size for decision trees or depth for neural nets with a fixed architecture. We assume that $\mathcal{F}$ contains all constant functions and that $\mu(c) = 0$ for any constant function $c$.

- We consider post-processing functions $\kappa : [0,1] \to [0,1]$ belonging to some family $K$, which have bounded complexity under $\mu$. Formally, there exists $b = b(K) \in \mathbb{Z}$ such that for every $f \in \mathcal{F}$ and $\kappa \in K$,
$$\mu(\kappa \circ f) \leq \mu(f) + b$$
.

- We have a loss function $\ell : [0,1] \times \mathbb{R} \to \mathbb{R}$. Let
$$\ell(f, \mathcal{D}) = \mathop{\mathbb{E}}_{(x,y) \sim \mathcal{D}} [\ell(y, f)].$$

For any complexity bound $s$, we have an efficient procedure $\mathsf{Min}_{\mathcal{D}}(s)$ that can solve the loss minimization problem over models of complexity $s$:
$$\min_{\substack{f \in \mathcal{F} \\ \mu(f) \leq s}} \ell(f, \mathcal{D}).$$

The running time can grow with $s$. For each $s \in \mathbb{Z}$, let $\mathrm{OPT}_s$ denote the minimum of this program and let $f_s^*$ denote the model found by $\mathsf{Min}_{\mathcal{D}}(s)$ that achieves this minimum.

We use this to present an algorithm that produces a model $f$ which is smoothly calibrated, has loss bounded by $\mathrm{OPT}_s$, and complexity not too much larger than $s$.

---

**Algorithm 1** Local search for small pGap.

---

Input $s_0$.
$s \leftarrow s_0$.
$h \leftarrow 0$.
**while** True **do**
    $f_s^* = \mathsf{Min}(s)$.
    $f_{s+b}^* = \mathsf{Min}(s + b)$.
    **if** $\mathrm{OPT}_{s+b} \geq \mathrm{OPT}_s - \alpha$ **then**
        Return $f_s^*$.
    **else**
        $h \leftarrow h + 1$.
        $s \leftarrow s + b$.
    **end if**
**end while**

---

**Theorem G.1.** *On input $s_0$, Algorithm 1 terminates in $h \leq 1/\alpha$ steps and returns a model $f \in \mathcal{F}$ which satisfies the following guarantees:*

1. $\mu(f) = t \leq s_0 + hb$.

2. $\ell_2(f, \mathcal{D}) = \mathrm{OPT}_t \leq \mathrm{OPT}_{s_0} - h\alpha$.

3. $\mathsf{pGap}(f) \leq \alpha$, *hence* $\mathsf{smCE}(f) \leq \sqrt{\alpha}$.

*Proof.* The bound on the number of steps follows by observing that each update decreases $\ell_2(f, \mathcal{D})$ by $\alpha$. Since $\mathrm{OPT}_{s_0} \leq \mathrm{OPT}_0 \leq 1$, this can only happen $1/\alpha$ times. This also implies items (1) and (2), since each update increases the complexity by no more than $b$, while decreasing the loss by at least $\alpha$.

To prove claim (3), assume for contradiction that $\mathsf{pGap}(f) > \alpha$. Then there exists $\kappa \in K$ such that if we consider $f' = \kappa \circ f$, then
$$\mu(f') \leq t + b, \quad \ell_2(f, \mathcal{D}) < \mathrm{OPT}_t - \alpha.$$
But this $f'$ provides a certificate that
$$\mathrm{OPT}_{t+b} \leq \ell_2(f', \mathcal{D}) < \mathrm{OPT}_t - \alpha.$$
Hence the termination condition is not satisfied for $f$. So by contradiction, it must hold that $\mathsf{pGap}(f) \leq \alpha$, hence $\mathsf{smCE}(f) \leq \sqrt{\alpha}$. $\qquad\square$

We now present an alternative algorithm that is close in spirit to algorithm 1, but can be viewed more directly as minimizing a regularized loss function. We assume access to the same minmization procedure $\mathsf{Min}_{\mathcal{D}}(s)$ as before. We define the regularized loss

$$\ell^\lambda(f, \mathcal{D}) = \ell(f, \mathcal{D}) + \lambda\mu(f).$$

$\lambda$ quantifies the tradeoff between accuracy and model complexity that we accept. In other words, we accept increasing $\mu(f)$ by 1 if it results in a reduction of at least $\lambda$ in the loss.

---

**Algorithm 2** Regularized loss minimization for small pGap.

---

$\quad r \leftarrow \lfloor 1/\lambda \rfloor$.
$\quad$**while** $s \in \{0, \cdots, r\}$ **do**
$\quad\quad f_s^* \leftarrow \mathsf{Min}(s)$.
$\quad\quad \mathrm{OPT}_s \leftarrow \ell_2(f_s^*, \mathcal{D})$.
$\quad$**end while**
$\quad t \leftarrow \mathrm{argmin}_{s\in\{0,\ldots,r\}} \mathrm{OPT}_s + \lambda s$.
$\quad$Return $f_t^*$.

---

**Theorem G.2.** *Let $\lambda = \alpha/b$. Algorithm 2 returns a model $f$ with $\mu(f) = t \leq \lfloor 1/\lambda \rfloor$ where*

1. *$f = \mathrm{argmin}_{f'\in\mathcal{F}} \ell^\lambda(f', \mathcal{D})$ and $\ell_2(f, \mathcal{D}) = \mathrm{OPT}_t$.*

2. *For any $s \neq t$,*
$$\mathrm{OPT}_s \geq \mathrm{OPT}_t - (s-t)\lambda.$$

3. *$\mathsf{pGap}(f) \leq \alpha$, hence $\mathsf{smCE}(f) \leq \sqrt{\alpha}$.*

*Proof.* We claim that
$$\min_{f'\in\mathcal{F}} \ell^\lambda(f', \mathcal{D}) \leq 1.$$

This follows by taking the constant $1/2$, which results in $\ell_2(0, \mathcal{D}) \leq 1$ and $\mu(f) = 0$. This means that the optimum is achieved for $f$ where $\mu(f) \leq 1/\lambda$. For $f$ such that $\mu(f) \leq t$, it is easy to see that Algorithm 2 finds the best model.

Assume that $s \neq t$. By item (1), we must have
$$\mathrm{OPT}_t + \lambda t \leq \mathrm{OPT}_s + \lambda s$$
Rearranging this gives the claimed bound.

Assume that $\mathsf{pGap}(f) = \alpha' > \alpha$. Then there exists $f' = \kappa(f)$ such that $\mu(f') \leq \mu(f) + b$, but $\ell(f', \mathcal{D}) \leq \ell_2(f, \mathcal{D}) - \alpha'$. But this means that
$$\begin{aligned}
\ell^\lambda(f') &= \ell(f', \mathcal{D}) + \lambda\mu(f') \\
&\leq \ell_2(f, \mathcal{D}) - \alpha' + \lambda(\mu(f) + b) \\
&= \ell^\lambda(f) + (\lambda b - \alpha') \\
&< \ell^\lambda(f)
\end{aligned}$$
where the inequality holds by our choice of $\lambda = \alpha/b$. But this contradicts item (1). $\quad\square$

Note that if we take $s - t = hb$, then $(s-t)\lambda = h\alpha$, so Item (2) in Theorem G.2 gives a statement matching Item (2) in Theorem G.1. The difference is that we now get a bound for any $s \neq t$. When $s > t$, item (3) upper bounds $\mathrm{OPT}_t - \mathrm{OPT}_s$, which measures how much the $\ell_2$ loss decreases with increased complexity. When $s < t$, $\mathrm{OPT}_s > \mathrm{OPT}_t$, and item (3) lower bounds how much the loss increases when shrink the complexity of the model.

**Discussion: Importance of depth.** These two algorithms highlight the central role of *composition* in function families. Specifically, if we are optimizing over a function family that is not closed under univariate Lipshitz compositions (perhaps approximately), then we have no reason to expect $\mathsf{pGap}(f) \approx 0$: because there could exist a simple (univariate, Lipshitz) transformation which reduces the loss, but which our function family cannot model. This may provide intuition for why certain models which preceded DNNs, like logistic regression and SVMs, were often not calibrated. Moreover, it emphasizes the importance of depth in neural-networks: depth is required to model composing Lipshitz transforms, which improves both loss and calibration.

## G.2 Heuristic Structure of SGD

Finally, there is an informal conjecture which could explain why certain DNNs in practice are calibrated, even if they are far from state-of-the-art: Informally, if it were possible to improve the loss by a simple update to the network function (post-processing), then one may conjecture SGD would have exploited this by the end of training. That is, updating the parameters from computing the function $f$ to computing $(\kappa \circ f)$ may be a "simple" update for SGD on a deep-enough network, and SGD would not leave this gain on the table by the end of training. We stress that this conjecture is informal and speculative, but we consider it a plausible intuition.

# H Helper Lemmas

**Claim H.1.** *Let $\eta_1, \eta_2 : [0,1] \to \mathbb{R}$ be two 1-Lipschitz functions. Then $\max(\eta_1(v), \eta_2(v))$ and $\min(\eta_1(v), \eta_2(v))$ are also 1-Lipschitz functions of $v \in [0,1]$.*

*Proof.* Fix arbitrary $v, v' \in [0,1]$. Let $i, i' \in \{1,2\}$ be such that $\max(\eta_1(v), \eta_2(v)) = \eta_i(v)$ and $\max(\eta_1(v'), \eta_2(v')) = \eta_{i'}(v')$. We have

$$\max(\eta_1(v), \eta_2(v)) - \max(\eta_1(v'), \eta_2(v')) \geq \eta_{i'}(v) - \eta_{i'}(v') \geq -|v - v'|, \quad \text{and}$$
$$\max(\eta_1(v), \eta_2(v)) - \max(\eta_1(v'), \eta_2(v')) \leq \eta_i(v) - \eta_i(v') \leq |v - v'|.$$

This proves that $\max(\eta_1(v), \eta_2(v))$ is 1-Lipschitz. A similar argument works for $\min(\eta_1(v), \eta_2(v))$.
$\square$

**Lemma H.2.** *Let $\eta : [0,1] \to [-1,1]$ be a 1-Lipschitz function. Define $\kappa : [0,1] \to [0,1]$ such that $\kappa(v) = \mathsf{proj}_{[0,1]}(v + \eta(v))$ for every $v \in [0,1]$. Define $\eta' : [0,1] \to [-1,1]$ such that $\eta'(v) = \kappa(v) - v$ for every $v \in [0,1]$. Then $\eta'$ is 1-Lipschitz.*

*Proof.* By the definition of $\kappa$, we have $\kappa(v) = \min(\max(v + \eta(v), 0), 1)$. Therefore, $\eta'(v) = \min(\max(\eta(v), -v), 1 - v)$. The lemma is proved by applying Claim H.1 twice. $\square$

**Lemma H.3.** *Let $V \subseteq [a,b]$ be a non-empty set. For some $m \in \mathbb{R}$, let $\varphi : V \to [m, +\infty)$ be a function lower bounded by $m$. Define $\psi : \mathbb{R} \to \mathbb{R}$ such that $\psi(t) = \sup_{v \in V} \left( vt - \varphi(v) \right)$. Then $\psi$ is a convex function. Moreover, for $t_1 < t_2$, we have*

$$a(t_2 - t_1) \leq \psi(t_2) - \psi(t_1) \leq b(t_2 - t_1).$$

*Proof.* For any $p \in [0,1], t_1, t_2 \in \mathbb{R}$, and $v \in V$,

$$v(pt_1 + (1-p)t_2) - \varphi(v) = p(vt_1 - \varphi(v)) + (1-p)(vt_2 - \varphi(v)) \leq p\psi(t_1) + (1-p)\psi(t_2).$$

Taking supremum over $v \in V$, we have

$$\psi(pt_1 + (1-p)t_2) \leq p\psi(t_1) + (1-p)\psi(t_2).$$

This proves the convexity of $\psi$. Moreover, if $t_1, t_2 \in \mathbb{R}$ satisfies $t_1 < t_2$, then for any $v \in V$,

$$vt_1 - \varphi(v) = vt_2 - \varphi(v) + v(t_1 - t_2) \leq vt_2 - \varphi(v) + a(t_1 - t_2) \leq \psi(t_2) - a(t_2 - t_1).$$

Taking supremum over $v \in V$, we have

$$\psi(t_1) \leq \psi(t_2) + a(t_1 - t_2).$$

Similarly,

$$vt_2 - \varphi(v) = vt_1 - \varphi(v) + v(t_2 - t_1) \leq vt_1 - \varphi(v) + b(t_2 - t_1) \leq \psi(t_1) + b(t_2 - t_1).$$

Taking supremum over $v \in V$, we have

$$\psi(t_2) \leq \psi(t_1) + b(t_2 - t_1).$$
$\square$

