# OpenReview forum: "When Does Optimizing a Proper Loss Yield Calibration?"
_NeurIPS.cc/2023/Conference — NeurIPS 2023 spotlight_

### Official Review · Reviewer_CHsP · 2023-06-19

**Soundness:** 4 excellent
**Presentation:** 4 excellent
**Contribution:** 4 excellent
**Rating:** 9
**Confidence:** 3

**Summary:**

This paper elucidates the relationship between proper losses and calibration by providing the minimal necessary and sufficient condition for proper losses to induce calibrated models.
The condition, local optimality, delineates the concept that no post-processing functions can improve a proper loss anymore.
This is related to a specific measure of calibration called smooth calibration error, which is a kind of correlation between miscalibration and model predictions.
The smooth calibration error is used here because it does not suffer from a discontinuous nature, unlike the popular expected calibration error, and it naturally emerges from the Bregman divergence structure.
By leveraging the Bregman divergence structure of proper losses and convex analysis, a theoretical connection between the smooth calibration error and the post-processing gap (the quantitative version of the local optimality condition) is established, which indicates that minimizing the post-processing gap is necessary and sufficient to achieve sufficiently small calibration error.
As an implication of the theory, the authors point out a potential connection between the implicit bias regularization of DNNs and the post-processing gap so that sufficiently over-parametrized networks may achieve a small post-processing gap, leading to the current well-calibrated neural networks.
Overall, this paper pushes the understanding of proper losses and calibration toward the context of modern neural network regimes.

**Strengths:**

- A new connection between proper losses and calibration: Though the two concepts seek similar goals, they have been studied independently, and the relationship has not been understood well. The main theorem of this paper draws the connection by establishing the upper and lower bounds of the post-processing gap (related to proper losses) by the smooth calibration error (related to calibration). As far as I know, this is the first attempt to connect the two concepts. The concept of the post-processing gap is well motivated by deep neural networks (regarding the post-process as the layer addition).
- A transparent proof: The proof of the main result (Theorem E.8) gives us a nice picture of the relationship between the calibration error and proper losses. Specifically, the proof of the bounds mostly leverages the smoothness of a function $\\psi$. This proof is simple and gives us an insight that the structure of $\\psi$ essentially governs the connection.
- The clarity of the presentation: Though the concepts introduced in this paper are rather dense, the authors did a nice job of presenting them gradually from a conceptual level to a technical level, which helps readers who may not be familiar with those concepts understand them easily.

**Weaknesses:**

- Potential gaps between the theory and Guo et al. (2017): The authors argue that "the previous generation of image classifiers were poorly calibrated [Guo et al., 2017]" and "state-of-the-art DNNs are often well-calibrated: because their test loss cannot be improved much by adding a few more layers." However, I feel that the architectures used by Guo et al. (2017) are already very deep. For example, in their pilot study (Figure 1), they chose to use a 110-layer ResNet, which would be sufficiently over-parametrized. Since Guo et al. (2017) argued that DNNs are poorly calibrated even with that number of layers, I would like to see the authors' discussion on this line.
- Missing key references: Some results and claims in the paper are substantially related to previous works that are not mentioned in the paper. It would be great to give credit to them. For example, the dual mapping of the form $\\mathsf{dual}(v) := \\ell(0,v) - \\ell(1,v)$ (l. 299) is known in Eq. (47) of Buja et al. (2005); The composition of a proper loss $\\ell$ and the dual mapping $\\psi$ is known as composite losses, coined in Reid and Williamson (2010). The dual loss form in Eq. (6) and Definition 4.3 is known as the Fenchel-Young losses, and the expression was shown in Eq. (14) of Blondel et al. (2020) and Eq. (11) of Duchi et al. (2018); The convex conjugate structure $\\nabla\\psi(\\mathsf{dual}(v)) = v$ (l. 329) was pointed out in Figure 1 of Bao and Sugiyama (2021); Some parts of Lemma E.4 are closely related to Proposition 2 of Blondel et al. (2020).
- Restriction to binary classification: The attention of this whole paper is restricted to binary classification, as mentioned in the conclusion. This is far more restrictive in practical situations. But I don't think this limitation undermines the impact of this paper.


**References**

- Reid and Williamson (2010) "Composite Binary Losses"
- Buja et al. (2005) "Loss Functions for Binary Class Probability Estimation and Classification: Structure and Applications"
- Blondel et al. (2020) "Learning with Fenchel-Young Losses"
- Duchi et al. (2018) "Multiclass Classification, Information, Divergence, and Surrogate Risk"
- Bao and Sugiyama (2021) "Fenchel-Young Losses with Skewed Entropies for Class-posterior Probability Estimation'

**Questions:**

Some minor comments:

- (Question) At l.261, I don't get the point of "within the restricted class where the logit is a linear combination of the features," specifically, what "the features" mean in the current context. Could you clarify?
- (Question) Why do you present the non-generalized dual calibration error in Section 4, unlike the generalized one in the appendix? I do not get the reason why it is useful to confine the generalization $w(x,g(x))$ to $\\eta(x)$.
- (Typo) At l.75, "depends on on the architecture" -> "depends on the architecture"
- (Typo) At l.204, a parenthesis is missing for $\\eta(f(x))$.

**Limitations:**

The authors discuss the limitations of this work at the end of the paper and point out that there is room to investigate the connection among calibration, DNN architectures, and optimization.

Most of the work is theoretical, and few negative societal concerns would apply.

---

> ### Author Rebuttal · Authors · 2023-08-09
>
> Thank you for the positive feedback! Here is our response to the individual comments and questions in your review:
>
> **Re. relation to Guo et al. (2017).**
> Thanks for the great question. The issue is subtle. We discussed it in our paper (see Line 119) and would like to further elaborate below.
>
> It is simultaneously true that:
> 1. The DNNs used in [Guo et. al] are "deep enough"
> 2. The DNNs used in [Guo et. al] can be substantially improved (w.r.t test loss), by simple post-processing.
>
> The resolution is that the DNNs of [Guo et al.] were optimized for test *Error* (not test loss), and thus were  "overtrained" – they trained these DNNs for many epochs, enough to nearly interpolate the train set. This causes a high test loss, which they notice in Figure 3 of [Guo et al.]
> The authors themselves recognize this discrepancy between error and loss: "neural networks can overfit to NLL without overfitting to the 0/1 loss."
>
> In fact, if one trains the same DNNs as Guo et al, but optimizing for test *loss* (and thus, early-stopping before the loss overfits), then the resulting networks are very well-calibrated, consistent with our theory. (We have confirmed this experimentally, and given your question, we will consider adding this experiment to the full version).
>
> Our comment about "state-of-the-art DNNs are often well-calibrated" holds because, these days, DNNs are trained in the "very large data" regime, sometimes even for just 1 epoch — so, they have a small loss-generalization gap, and don't overfit their test loss.
> (e.g. the GPT-3 tech report which includes epoch details: https://arxiv.org/abs/2005.14165 , and Figure 16 of the GPT scaling laws paper [https://arxiv.org/abs/2001.08361],  which shows plots of test vs. train loss in both early-stopped and overfitting regimes).
>
> To rephrase, the essential difference between SOTA networks in 2017 and 2023 is: In 2017, classification networks were trained for many epochs, and thus had very high test loss (despite small test error). But in 2023, we often have so much data that we cannot afford many epochs, and thus the test loss does not overfit.
>
> We can add this more extended discussion to the final version of our paper.
>
> **Re. missing references.** We are grateful that you point us to these references! They are very relevant and we will make sure to cite them properly in our final version. We stated in our submission that the connection between proper losses and conjugate pairs of convex functions is well known before our work. We are eager to include a more comprehensive discussion about where each specific notion/result has appeared in the literature. Your suggested references are extremely helpful for us to this end.
>
> **Re. beyond binary classification.** See our response to reviewer ooFk re. multiclass settings.
>
> **Re. features in logistic regression.** We are referring to the standard logistic regression setting where each data point (x,y) has a feature vector $x\\in \\mathbb R^d$ and a binary label $y\\in \\{0,1\\}$. Given a number of such data points, our goal is to learn a linear/affine function $g(x) = \\langle a,x\\rangle + b$ that maps a feature vector x to a logit g(x), after which we apply the sigmoid transformation $\\sigma$ to get the final prediction $\\sigma(g(x))\in [0,1]$. Here the logit g(x) is restricted to the form $g(x) = \\langle a,x\\rangle + b$. That is, the logit is a linear combination of the coordinates of x plus a bias b. Each coordinate of the feature vector x is usually called a feature, so the logit is a linear combination of the features plus a bias. We will add “plus a bias” to our paper to be more accurate.
>
> **Re. presentation of generalized dual calibration error.** We choose to defer our results about generalized dual calibration error to the appendix because 1) we have limited space, and 2) we do not want our core calibration results to be diluted by the more general results. In our final version, we will add text to our main paper mentioning and pointing to the more general results in the appendices so that interested readers will not miss them.

---

> > ### Comment · Reviewer_CHsP · 2023-08-14
> > **Response**
> >
> > I thank the authors for addressing my concerns. Specifically, the clarification on the relation to Guo et al. (2017) was helpful. It seems to become less misleading to add such a discussion in the next version to clarify the difference of the practice of neural networks training during these few years.

---

### Official Review · Reviewer_5Dzr · 2023-06-26

**Soundness:** 4 excellent
**Presentation:** 3 good
**Contribution:** 3 good
**Rating:** 7
**Confidence:** 4

**Summary:**

This work introduces a local optimality condition for models (with respect to proper losses) based on (additive) post-processing with a 1-Lipschitz function that is necessary and sufficient for calibration. The authors also connect their results to the idea of implicit regularization, showing that structural risk minimization with an appropriate class of complexity measures achieves good calibration.

**Strengths:**

## Originality
The introduced notion of post-processing error and its connection to (smooth) calibration is novel and useful, as it proves plausible explanations for why current SOTA deep learning models are better calibrated when compared to previous generations. Additionally, the introduced local optimality condition based on post-processing is distinct from conditions seen in optimization, but remains non-trivial since the class of post-processing functions is restricted to be 1-Lipschitz.

## Quality
The paper is technically sound and polished; the definitions are well-motivated and simpler results are accompanied with proofs (or at least intuitive justification) in the main body of the paper.

## Clarity
The theory is easy to follow, and the authors have done a good job of scaling the complexity of the results as the paper progresses (introducing simple examples and high-level ideas first, and generalizing later). The authors also qualitatively connect their results to recent empirical phenomena in deep learning, which is helpful for grounding the theory.

## Significance
Calibration is an increasingly important problem, and the paper provides insights into how to think about the relationship between calibration and post-processing procedures.


**Weaknesses:**

- *Experiments:* While I understand this is a theory paper, I do think it would be nice to have some experimental analysis of whether the local optimality condition is satisfied for modern architectures (i.e. simply adding an extra layer as suggested in the paper and analyzing calibration performance).
- *Addressing Popular Post-Processing Methods:* If I understand the discussion around Definition 2.2 correctly, the post-processing operation defined does not include temperature-scaling-type techniques, since $f(x)$ includes the sigmoid operation and one would have to apply the re-scaling to the logits. If this is not the case, some clarification in this section linking the post-processing definition back to the standard post-processing approaches would be useful.

**Questions:**

My only question is related to my discussion under weaknesses; namely how temperature scaling type methods would fit in with Definition 2.2.

**Limitations:**

The authors appropriately address limitations; namely, they acknowledge that their results do not explain _why_ the optimization process for current models leads to calibrated predictors.

---

> ### Author Rebuttal · Authors · 2023-08-09
>
> Thank you for the positive feedback! Here is our response to the individual comments and questions in your review:
>
> **Re. temperature scaling.** Typically temperature scaling is applied to the logits when we optimize the cross entropy loss. Thus temperature scaling is more closely related to Definition 2.5 (which uses the cross-entropy loss) instead of Definition 2.2 (which uses the squared loss). In Definition 2.5 we apply a Lipschitz post-processing $\\kappa$ to the **logit** g(x) (see text around Line 225). Temperature scaling, i.e., dividing the logit g(x) by a temperature parameter T > 0, is a special case of such Lipschitz post-processings as long as T is not too close to zero (we need this assumption on T to ensure Lipschitzness). Definition 2.5 also considers many other Lipschitz post-processings that cannot be expressed as temperature scaling. This is important because our logistic regression example in Appendix B shows that a small post-processing gap w.r.t temperature scalings alone does NOT imply a small calibration error.
>
> We will include this discussion about temperature scaling in our final version.
>
> **Re. experiments.** See our response to Reviewer 6rky re. experiments.

---

> > ### Comment · Reviewer_5Dzr · 2023-08-12
> >
> > Ah, thank you - that makes sense. I have no further questions.

---

> > > ### Comment · Area_Chair_tGyD · 2023-08-12
> > > **"Sense" or "no sense"**
> > >
> > > Dear Reviewer, did you mean to say "That makes sense" or "That makes no sense"? In the case of the second, can you please specify what in particular? Thanks :).

---

> > > > ### Comment · Reviewer_5Dzr · 2023-08-16
> > > >
> > > > Sorry for the typo, I meant "makes sense". My apologies for the delayed response.

---

### Official Review · Reviewer_ooFk · 2023-07-06

**Soundness:** 3 good
**Presentation:** 3 good
**Contribution:** 3 good
**Rating:** 4
**Confidence:** 3

**Summary:**

The paper considers calibration in binary classification when training was performed with proper losses. The authors showed that
the post-processing gap of a predictor, which is a maximum improvement of the loss given any 1-Lipschitz update (calibration) function,
could be both lower and upper bounded by a simple expression of smooth calibration error. Moreover, the authors proved that the constants used in upper and lower bounds are optimal.

**Strengths:**

The paper describes the connection between proper loss landscapes and calibration error improvement, which is an important and quite original problem under the current formulation.

The paper has good positioning among other papers in the field, with clear references to many studies and a detailed indication of what was their contribution.

The clarity of the paper is good.

**Weaknesses:**

The link with potential practical application is missing (or rather not convincing). Adding small-scale experiments to support theoretical results would be beneficial, e.g. to check whether the pre-requisites of the theoretical results are (approximately) fulfilled in practice.

The discussion and explanation of theoretical results could be improved. The current impression is that either theoretical results are not that impressive (and based mostly on properties of Bregman divergence) or that the potential impact of derived theory is oversold.


**Questions:**

1. The derived results are valid in binary classification settings. Will it be possible, and what are the possible limitations, to obtain similar results in multiclass settings?

2. Could it be said that all theoretical derivations are based on properties of Bregman divergences but in a new setting connected with calibration errors?


**Limitations:**

The limitations have been covered at the end of the paper.

No need to discuss societal impact in this paper.

---

> ### Author Rebuttal · Authors · 2023-08-09
>
> We thank the reviewer for their thoughtful feedback, and for recognizing our work on an "important and quite original problem." We are glad the reviewer rates our soundness, presentation, and contribution as "good." We are thus unsure why the overall score is a borderline reject, but we have included our response to the individual comments and questions below. If the reviewer is satisfied with our responses, and believes this paper should appear in NeurIPS, we ask that they consider increasing their score to an Accept.
>
> **Re. practical applications.** Our theory can be viewed as a bridge connecting calibration and proper loss minimization. Calibration is a desired property in numerous practical applications, but our understanding of how to achieve calibration is generally not as profound as how to perform loss minimization. Our theory allows people to apply the machinery of loss minimization to reason about calibration, and we hope that our theory will inspire better principled approaches to calibration.
>
> Here is an immediate impact of our paper to practical application. Practitioners (using sklearn for instance) are currently encouraged to believe that solely by minimizing proper loss within a restricted family of functions, they will get calibrated predictors (see the references in our paper). We show convincingly that in this generality, the claim is false. Moreover, we identify a sufficient and necessary local optimality condition for proper loss minimization that guarantees calibration. By being aware of our corrected connection between loss minimization and calibration, practitioners can avoid false confidence in a model’s calibration and eventually build more calibrated models.
>
> **Re. multiclass settings.** Our results can be extended to the multiclass setting, but such an extension is not entirely straightforward. Even the definition of calibration in the multiclass setting is subtle. In the multiclass setting, the prediction f(x) becomes a vector that assigns a probability to each class. One way to define calibration is to condition on the entire vector f(x) (canonical calibration), whereas other ways may involve conditioning on certain coordinates of f(x) (e.g. confidence calibration conditions on the maximum coordinate). How to robustly and efficiently measure the distance to each type of calibration in the multiclass setting adds another layer of complexity, and the recent work [Błasiok et al., 2023] only answers this question for binary labels. Our theory is more easily extended to canonical calibration than confidence calibration, but confidence calibration is widely used in practice. We thus leave the important but challenging question of building a complete extension of our theory in the multiclass setting to future work. We believe this is more conducive to the community than to include in the current paper an incomplete theory for the multiclass setting that may risk being misleading.
>
> **Re. properties of Bregman divergence.** Yes, as we mention in our submission, a lot of our proof techniques are closely related to properties of the Bregman divergence and the theory of convex conjugates. A major part of our contribution is formulating the right theorems to prove, which is as important as proving those theorems. We believe that by presenting our proof techniques in such a way that they appear relatively simple in retrospect, we make our paper more easily digestible and thus potentially more impactful.
>
> **Re. experiments.** See our response to Reviewer 6rky re. experiments.

---

> > ### Comment · Reviewer_ooFk · 2023-08-16
> > **Re: rebuttal**
> >
> > Thanks for the clarifications!
> >
> > **Re: Re: practical applications.** You have written in your response that _"Practitioners (using sklearn for instance) are currently encouraged to believe that solely by minimizing proper loss within a restricted family of functions, they will get calibrated predictors (see the references in our paper)"_. I am not convinced that it is true. Your work refers to a paper published in 2011 about sklearn documentation. Back then, there might indeed have been such a belief about proper losses, given that no large NNs models existed. The up-to-date sklearn documentation about calibration does not seem to contain such encouragement. In my opinion, it is common knowledge that models tend to be uncalibrated, unless post-hoc calibration is used.
> >
> > **Re: Re: experiments.** Please comment on the two following points and questions.
> >
> > 1. Experimental evidence you are referring to is mainly about models trained with cross-entropy. However, your theoretical results are valid for all proper losses. Could we expect that new large NNs could be trained basically with any proper loss and have a nearly perfect calibration, given, for example, a slightly larger dataset/architecture, than existing models trained with cross-entropy have?
> >
> > 2. You claimed that your theoretical results provide an explanation of calibration differences between an old and new generation of NN models (due to a much larger dataset and more complex architecture). Those calibration differences are the main experimental evidence of your theory in the paper.
> > However, it could probably be explained in a simpler way: proper losses should have well-calibrated results on a train set. Once train set is big enough (as for new generation of NN models), we could expect that train set empirical distribution is very close to ground truth population distribution, so the results on the test set (generated from this unknown population distribution) would be well-calibrated.
> > Could you comment on this reasoning, and do you agree with it? Could you suggest other existing experimental evidence that supports your theoretical results as opposed to the above simpler explanation?

---

> > > ### Author Response · Authors · 2023-08-16
> > >
> > > Thank you for replying!
> > >
> > > **Re: practical applications.**
> > > We believe our quoted statement is well-justified for the following reasons:
> > >
> > > First, our statement is not just about neural networks, it is about minimizing proper losses *within a restricted family of functions.* Logistic regression is an instance of such restricted minimization, and many sources claim logistic regression is well-calibrated primarily because it minimizes a proper loss. As mentioned in our paper, the sklearn documentation as of August 16 2023 says "LogisticRegression returns well calibrated predictions by default as it has a canonical link function for its loss, i.e. the logit-link for the Log loss" (from the url we cited: https://scikit-learn.org/stable/modules/calibration.html). **Our 2011 citation is for the Scikit-learn package itself, and we also cited the up-to-date documentation url.** The wikipedia page on Platt scaling (as of August 16 2023) also claims that logistic regression gives well-calibrated models: “(Platt scaling) has less of an effect with **well-calibrated models such as logistic regression**, …” (https://en.wikipedia.org/wiki/Platt_scaling).
> > >
> > > However, our analysis demonstrates that this is not true: logistic regression can be severely mis-calibrated, essentially because it is not minimizing over a sufficiently rich family of functions (see lines 42-50, and Appendix B of our paper).
> > > Logistic regression remains an extremely popular method among practitioners of machine learning and data science in the real world, even in this age of "large NNs", and thus we believe our statement on practical relevance is justified.
> > >
> > > Finally, regarding neural networks, we do not agree with the claim that "it is common knowledge that models tend to be uncalibrated, unless post-hoc calibration is used." Specifically, because it is not true: there are now many documented instances of modern neural networks being extremely well-calibrated out-of-the box, as we cited in our paper. These citations include, for example, the GPT-4 tech report (Figure 8, left, in: https://arxiv.org/abs/2303.08774).
> > >
> > > **Re: experiments.**
> > > 1. Yes, our theory does predict that minimizing the test loss for essentially any proper loss over a rich enough model family will yield a calibrated model. We observed this experimentally in an ongoing and upcoming work.
> > > 2. Regarding the potential simpler explanation: we agree with the logic of the explanation (see our response to Reviewer CHsP) and the explanation was proposed by Carrell et al., 2022 (cited in our paper). However, the premise that “proper losses should have well-calibrated results on a train set” is non-trivial to establish and it is a motivation of our work. This premise is not true in general: logistic regression can be poorly calibrated on its train set, despite minimizing a proper loss. It turns out that sufficiently large neural networks *are* in fact well-calibrated on their train sets, as observed experimentally by Carrell et al., 2022, and these observations were inspiration for our theoretical work.

---

> > > > ### Comment · Reviewer_ooFk · 2023-08-17
> > > >
> > > > First, the claim “(Platt scaling) has less of an effect with well-calibrated models such as logistic regression, …” from Wikipedia should not be interpreted as claiming some sort of a guarantee. It is obvious that (nearly) every model is miscalibrated since the requirement to be calibrated requires equality of real values which is almost surely not satisfied. The only question is how much data are needed to demonstrate miscalibration. Therefore, the claim of logistic regression being well-calibrated is an empirical one - in many practical applications people have observed that logistic regression tends to be relatively well-calibrated. But as you state in your Appendix B lines 536-537, "Similar examples of logistic regression being miscalibrated are found in Kull et al. [2017]." Note that your mentioned lines 42-50 give a false impression as if this 'discovery' were your novel contribution.
> > > >
> > > > Second, regarding your claim that "there are now many documented instances of modern neural networks being extremely well-calibrated out-of-the box" and your subsequent reference to the GPT-4 tech report. The report itself is much more modest by stating "Interestingly, the pre-trained model is highly calibrated (its predicted confidence in an answer generally matches the probability of being correct)." They don't seem to have run any statistical hypothesis test on this, so it might even be that the model is significantly miscalibrated. I agree that it is more calibrated than some other networks that we have seen in the past years, but it might still benefit from further posthoc calibration (unless checked and proved otherwise). For example, the two rightmost columns in Table 1 of the NeurIPS paper (Mukhoti et al 2020) https://papers.neurips.cc/paper/2020/file/aeb7b30ef1d024a76f21a1d40e30c302-Paper.pdf
> > > > show that even though models trained with focal loss were shown in that paper to be surprisingly well-calibrated, temperature scaling with temperatures in the range from 0.9 to 1.1 still nearly always improved calibration further. This demonstrates that the model had still been miscalibrated, although to a much less extent than cross-entropy-trained models (keeping in mind that focal loss is not proper whereas cross-entropy is).
> > > >
> > > > Jishnu Mukhoti, Viveka Kulharia, Amartya Sanyal, Stuart Golodetz, Philip Torr, and Puneet Dokania. Calibrating deep neural networks using focal loss. Advances in Neural Information Processing Systems, 33:15288–15299, 2020.
> > > >
> > > > Therefore, I still stand by the view that it is common knowledge that models tend to be miscalibrated, unless post-hoc calibration is used. Miscalibration is here a continuum from mild to severe, but nearly always, post-hoc calibration has been shown to improve calibration, even if by just a small margin.
> > > >
> > > > Regarding the last point, I don't think you have responded to my question, "Could you suggest other existing experimental evidence that supports your theoretical results as opposed to the above simpler explanation?".

---

> > > > > ### Author Response · Authors · 2023-08-17
> > > > >
> > > > > We disagree on many points here, but this discussion on OpenReview does not appear to be converging to a resolution. We will thus leave our paper and existing comments to speak for themselves. We thank the reviewer for their time.

---

### Official Review · Reviewer_6rky · 2023-07-07

**Soundness:** 3 good
**Presentation:** 3 good
**Contribution:** 4 excellent
**Rating:** 7
**Confidence:** 3

**Summary:**

The paper provides a novel characterization of calibration as local optimality of the predictor w.r.t the global loss under post-processing of the prediction through a class of functions. The paper proves that any predictor satisfying such a condition is smoothly calibrated in the sense of [Kakade and Foster, 2008, Błasiok et al., 2023] and vice-versa. The paper further provides arguments for why such a condition should be satisfied by Deep Neural Networks (DNNs).

**Strengths:**

- The paper is well written, with a clear structure and ideas being developed in a coherent and logical sequence.
- The topic is of large importance in the present context of research in machine learning and has numerous implications for practical applications.
- The authors contribute a novel perspective on the 'implicit' calibration of machine learning models without the use of algorithms designed specifically for calibration, an aspect that is not extensively explored in current literature.
- The explicit characterization of calibration in terms of local optimality under post-processing transformations, while implicit in earlier works, appears to be novel. This equivalence could be useful for further theoretical analysis of calibration, especially for deep neural networks.
- The proposed framework is general and does not rely on specific choices of model's architecture or data-distribution.
- The paper provides partial explanations for the observed calibration of deep neural networks trained on large training datasets.

**Weaknesses:**

- **Limited technical contributions**: Claim 2.1 itself directly follows from the definition of perfect calibration. The main theoretical results in the paper are generalized formulations of Claim 2.1 to smooth calibration, lipschitz-post processing functions, and general proper losses. While these generalizations themselves are novel, their proofs involve straightforward algebra and convex analysis. Therefore, the technical and mathematical contributions of the paper are limited. The results can be strengthened from examples of non-trivial results that can be derived using the local-optimality based characterization of calibration. For deep neural networks, the present results are only suggestive and would benefit from additional details and concrete results.
- **Missing references**:
     - On double-descent in uncertainty quantification in overparametrized models: Lucas Clarte, Bruno Loureiro, Florent Krzakala, Lenka Zdeborova, Proceedings of The 26th International Conference on Artificial Intelligence and Statistics, PMLR 206:7089-7125, 2023.
     - Theoretical characterization of uncertainty in high-dimensional linear classification, Lucas Clarté, Bruno Loureiro, Florent Krzakala, and Lenka Zdeborová, Machine Learning: Science and Technology, Volume 4, Number 2

    The above papers analyze calibration for empirical risk minimization and like the present paper, also highlight the role played by
    regularization.

- **Dataset size and overparameterization**: The paper does not address aspects related to generalization and the effect of the training dataset size. While modern training setups utlize large training dataset sizes and one-pass SGD, their behavior in the proportional regimes of high-dimensional inputs and parameters
is non-trivial and not equivalent to the minimization of population loss. For instance, the above papers establish a double-descent like phenomenon for calibration for varying levels of overparametrization.

- **Experiments**: In light of the limited technical contributions, the paper could benefit from experimental justification of the validity of the local optimality condition for deep neural networks in realistic training setups.

**Questions:**

- What are some complexity measures satisfying the condition in Claim 4.8?
- What are the limitations of the definition of smooth calibration error used in the paper, especially in the context of deep neural networks?

**Limitations:**

Yes, the limitations have been adequately addressed. The work is theoretical in nature and does not have direct societal impacts.

---

> ### Author Rebuttal · Authors · 2023-08-09
>
> Thank you for the comprehensive review! Here is our response to the individual comments and questions in your review:
>
> **Re. technical contributions.** A main contribution of our work is formulating the right generalization of Claim 2.1 where 1) the notion of calibration is meaningful, and at the same time 2) the post-processings can be naturally implemented in deep neural networks that are not explicitly trained for calibration. We think these contributions should be weighed more than the technical difficulty of the proofs. Indeed, we think that the reason that such statements are not found in previous literature is because the right formulation is fairly subtle – for example, until recently it was not clear how to measure miscalibration in a reasonable way (e.g. ECE is not ideal as it's discontinuous; more in our response below re. the smooth calibration error). As an “example of non-trivial results that can be derived using the local-optimality based characterization of calibration”, our Claim 4.8 identifies the types of regularization that make structural risk minimization (SRM) produce well-calibrated predictors. A key step for establishing Claim 4.8 is using our theory to reduce the problem to determining what types of regularization make the solution to SRM satisfy the local optimality condition. In particular, our Claim 4.8 reveals that regularization plays a different role in *calibration* than it does for *generalization* – it is possible to have regularizers that work for both aspects, but it is not at all necessary.
>
> **Re. missing references and the generalization aspect.** Thank you for pointing us to these papers! In our final version, we will make sure to cite them and include a refined discussion about generalization based on them. As we mention in our submission, generalization concerns are out-of-scope for our paper – we consider only optimization aspects, by considering all quantities on the population distribution. Both generalization and optimization aspects are ultimately important to consider, but our paper focuses on only the latter.
>
> **Re. experiments.** Many existing experiments in the literature have verified that many neural networks trained via loss minimization exhibit good calibration performance. We mentioned many such examples in our submission. According to our theory, these neural networks must also satisfy the local optimality condition. That is, neural networks satisfying the local optimality condition in realistic training setups is a rather general phenomenon that has already been experimentally observed. Our contribution is drawing the connection between calibration and local optimality which gives an explanation for why previous experiments observe good calibration in models trained via loss minimization alone.
>
> **Re. complexity measures that satisfy the condition in Claim 4.8.** The size of a neural network, when multiplied by a suitable constant, satisfies the condition. This is because by the universality of neural networks, a 1-Lipschitz post-processing can be (approximately) implemented by adding a constant number of neurons to a neural network.
>
> **Re. limitations of smooth calibration error.** Recently Błasiok et al. [2023] showed that the smooth calibration error is a *consistent calibration measure*. Specifically, they showed that the smooth calibration error differs by at most a constant factor from the Wasserstein distance to perfect calibration. By definition, all consistent calibration measures are polynomially related to each other, so all our results about the smooth calibration error also hold for any consistent calibration measure, up to modifying constants in our inequalities. That is, by building our theory using the smooth calibration error, our results automatically apply to the kernel calibration error with the Laplace kernel, the interval calibration error (*modified* binned ECE with the right bin width and randomly shifted bins), and the distance to calibration itself. In contrast, the popularly used ECE and binned ECE are not consistent calibration measures and are not in general polynomially related to the distance to calibration. Błasiok et al. [2023] also showed efficient algorithms for estimating the smooth calibration error. Given the consistency, efficiency, and continuity of the smooth calibration error, perhaps a downside of it is that it is not currently used as often as the binned ECE, but this may change as people become more aware of the advantages of using a consistent calibration measure.
>
> **In conclusion:**
> If the reviewer is satisfied with our responses, and believes this paper should appear in NeurIPS, we ask that they consider increasing their score to an Accept.

---

> > ### Comment · Reviewer_6rky · 2023-08-15
> >
> > I thank the authors for providing clear and comprehensive responses. I've raised my rating on the assumption that the authors will incorporate discussions about the above points and missing references into the revised version of the paper.

---

### Official Review · Reviewer_PVPP · 2023-08-01

**Soundness:** 3 good
**Presentation:** 3 good
**Contribution:** 3 good
**Rating:** 7
**Confidence:** 3

**Summary:**

This paper seeks to explore and formalize the relationship between minimizing proper loss and calibration in machine learning models, particularly deep neural networks (DNNs). It presents a local optimality condition that is necessary and sufficient to ensure model calibration. The work discusses the implications of these findings on the calibration properties of modern DNNs and presents algorithms that can guarantee calibration. The paper also contrasts the differences in calibration between current and previous generation models through the lens of generalization.

**Strengths:**

NA

**Weaknesses:**

NA

**Questions:**

NA

---

> ### Author Rebuttal · Authors · 2023-08-09
>
> Thank you for the positive feedback!

---

### Decision · Program_Chairs · 2023-09-21

**Decision:**

Accept (spotlight)

**Comment:**

The contributions of this paper were appreciated in the review, and the paper was overall considered valuable for the community. The reviews presented many items towards improving the presentation of the paper which we trust the authors to take carefully into account as promised in the the discussion.